# Hypothermia evoked by stimulation of medial preoptic nucleus protects the brain in a mouse model of ischaemia

Shuai Zhang [1,3], Xinpei Zhang[1,3], Haolin Zhong[1], Xuanyi Li[1], Yujie Wu[1], Jun Ju [1], Bo Liu[1], Zhenyu Zhang[1], Hai Yan[1], Yizheng Wang[2], Kun Song [1] ✉ & Sheng-Tao Hou [1] ✉

Therapeutic hypothermia at 32-34 °C during or after cerebral ischaemia is neuroprotective. However, peripheral cold sensor-triggered hypothermia is ineffective and evokes vigorous counteractive shivering thermogenesis and complications that are difficult to tolerate in awake patients. Here, we show in mice that deep brain stimulation (DBS) of warm-sensitive neurones (WSNs) in the medial preoptic nucleus (MPN) produces tolerable hypothermia. In contrast to surface cooling-evoked hypothermia, DBS mice exhibit a torpor-like state without counteractive shivering. Like hypothermia evoked by chemogenetic activation of WSNs, DBS in free-moving mice elicits a rapid lowering of the core body temperature to 32-34 °C, which confers significant brain protection and motor function reservation. Mechanistically, activation of WSNs contributes to DBS-evoked hypothermia. Inhibition of WSNs prevents DBS-evoked hypothermia. Maintaining the core body temperature at normothermia during DBS abolishes DBS-mediated brain protection. Thus, the MPN is a DBS target to evoke tolerable therapeutic hypothermia for stroke treatment.

Current stroke therapies are inadequate, representing one of the most significant unmet medical needs[1]. The molecular cascades that set in after cerebral ischaemia are complex[2–5], which hamper the development of neuroprotective therapies. Hypothermia at mild to moderate levels (31–34 °C) during or following cerebral ischaemia is powerful neuroprotective[6–12]. Hypothermia can simultaneously inhibit multiple mechanisms of brain cell death and slow down metabolic processes to limit tissue damage[10,13,14], providing additive or synergistic beneficial clinical effects for stroke patients. The idea of using hypothermia for stroke therapy (therapeutic hypothermia) has been pursued for several decades[7,11,12]. However, currently used hypothermia triggered by peripheral cold sensors is ineffective in lowering the core body temperature ($T_{core}$) to the desired level[13,15]. The cooling process also evokes vigorous counteractive shivering thermogenesis that is difficult to tolerate in awake patients[16,17], and produces harmful side effects to the

heart, lungs, and brain[9]. A better method to induce therapeutic hypothermia is urgently needed.

A defining feature of mammalian and avian evolution is endothermy, achieved through the continuous homoeostatic regulation of the $T_{core}$ and metabolism[18–21]. However, many mammalian species, including mice, can initiate daily torpor and hibernation when faced with food shortages or harsh environmental conditions. The body enters into a hypothermic state, with the $T_{core}$ decreasing far below its homoeostatic set-point[18,21–23]. The thermoregulatory centre is in the preoptic area (POA) of the hypothalamus. Warm-sensitive neurones (WSNs) serve as a thermostat needed to achieve body cooling and hypothermia[18,21,24–28]. Several subpopulations of thermoregulatory neurones cluster together in the POA, and each is distinguished by a unique pattern of gene expression[21,29]. Excitatory signals from POA Vglut2-TRPM2 neurones, but not inhibitory Vgat neurones, promote

[1]Brain Research Centre, Department of Biology, School of Life Sciences, Southern University of Science and Technology, 1088 Xueyuan Blvd, Nanshan District, Shenzhen 518055 Guangdong, P.R. China. [2]Huashan Hospital, Fudan University, Shanghai, P.R. China. [3]These authors contributed equally: Shuai Zhang, Xinpei Zhang. ✉e-mail: songk@sustech.edu.cn; hou.st@sustech.edu.cn

hypothermia[27]. A population of glutamatergic *Adcyap1*-positive cells[21], Q-neurones[29] and oestrogen-sensitive medial preoptic area (MPA) neurones were also sufficient to drive a coordinated depression of metabolic rate and body temperature similar to torpor. Re-stimulation of WSNs that were activated during a previous bout of torpor is adequate to initiate the key features of torpor, even in mice that are not calorically restricted[21,30].

Torpor mice have reduced $T_{core}$, decreased movement, metabolic rate, sensory perception, and breathing without showing signs of shivering thermogenesis and organ damage[21,29,31]. It has, therefore, been suggested that inducing a torpor-like state in non-torpid species such as human patients experiencing brain trauma or surgery could provide a tool to reduce organ damage[32–34]. Based on these understandings of how torpor-like hypothermia is centrally evoked in endotherm animals, we hypothesised that deep brain stimulation (DBS) of preoptic WSNs could produce a torpor-like hypothermic state during experimental stroke without the occurrence of harmful side-effects of peripheral cold sensor-triggered hypothermia. The ultimate goal of the current study is to develop DBS-evoked hypothermia to benefit stroke patients.

Here, we show DBS-evoked hypothermia for brain protection against cerebral ischaemia in mice. Bilateral MPN DBS evokes a rapid and controllable level of mild to moderate hypothermia. DBS mice are awake, free-moving, and exhibit a torpor-like state without the occurrence of many adverse physiological and metabolic signs seen in

the peripheral cold-sensor-induced hypothermia. Activating preoptic WSNs by DBS lowers the thermoregulation "set-point", induces torpor-like hypothermia, limits tissue damage, and confers brain protection. The technique can be further developed to treat injured human brains known to benefit from hypothermia, such as stroke.

## Results

### WSNs-evoked hypothermia protects the ischaemic brain

We used a chemogenetics approach to specifically activate excitatory neurones (including WSNs) in the MPN to demonstrate that selective activation of WSNs triggers protective hypothermia. The expression of hM3DGq, a chemically activated receptor in Gq-DREADD (Gq-coupled Designer Receptor Exclusively Activated by Designer Drug), specifically in the excitatory Vglut2$^+$ preoptic WSNs, was used to trigger hypothermia. Song and colleagues[27] have previously demonstrated that systemic administration of chemical activator clozapine-N-oxide (CNO) into Vglut2-ires-Cre knock-in mice, pre-injected with adeno-associated viruses (AAVs) expressing Cre-dependent Gq-DREADD fused to mCherry (AAV-hSyn-DIO-Gq−mCherry) in the bilateral POA (Vglut2-Cre-CNO group), quickly induced hypothermia within 30 min to lower the $T_{core}$ stably to ~30 °C (Fig. 1a−c). Hypothermia lasted up to 12 h and could be induced repetitively without causing any apparent long-term adverse effects[27]. The $T_{core}$ returned to normothermia between 36.5-38 °C as CNO gradually wore off (Fig. 1b).

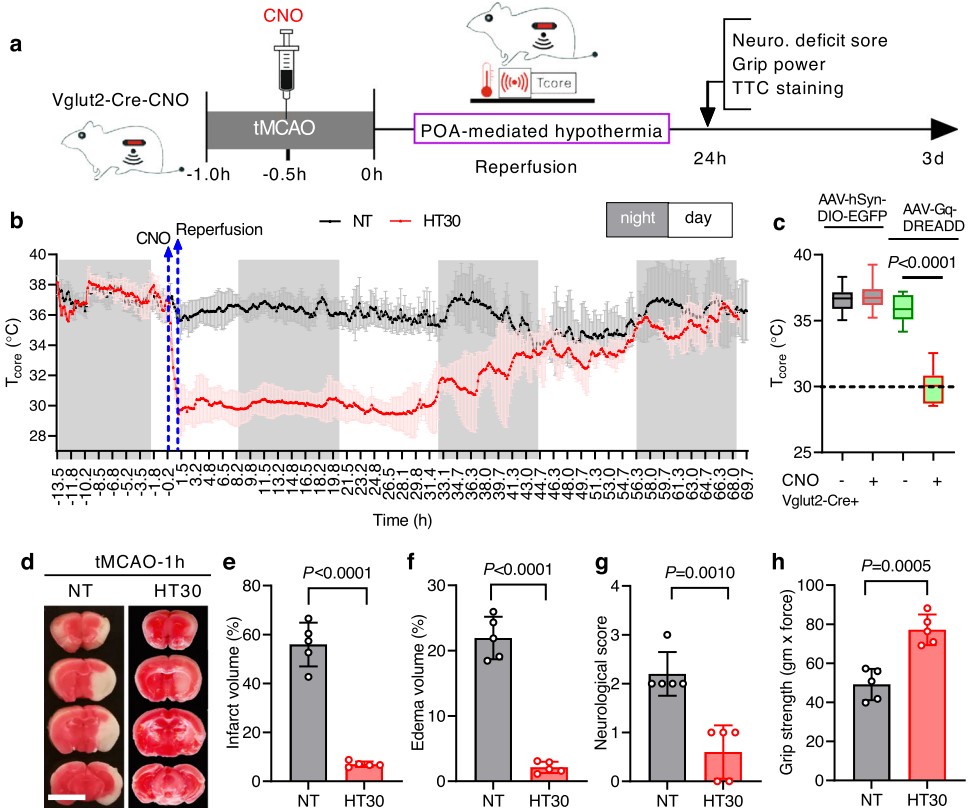

**Fig. 1 | Chemogenetic-evoked hypothermia protected the ischaemic brain during reperfusion. a** Schematic of experimental design. Vglut2-Cre mice were injected with control-AAV and Gq-DREADD-AAV for 3 weeks, followed by transient middle cerebral artery occlusion (tMCAO) surgery. **b** Clozapine-N-oxide (CNO) was injected via i.p. 1 h post tMCAO. $T_{core}$ reached 30 °C 1 h after CNO injection and stayed at 30 °C for almost 24 h during reperfusion. Recordings of $T_{core}$ were plotted against the 3 d reperfusion time. Grey-coloured boxes indicated the night-time, and the white area indicated the day-time. **c** The average $T_{core}$ 45 min before and after CNO injection was calculated and plotted. Box plots indicate median (middle line), 25th, 75th percentile (box) and the maximum and minimum values as endpoints for

the whiskers. One-way ANOVA (1ANOVA) with Sidak *post hoc* analysis, $F_{(4, 20)} = 35.24$, $P < 0.0001$; The specific $P$ value for comparison between groups was shown in the graph. **d** Brain coronal sections at 2 mm thickness were stained with 2,3,5-triphenyl tetrazolium chloride (TTC) to show viable tissue (red colour) and damaged tissue (white colour). Scale bar = 5 mm. **e**–**h** Unpaired two-tailed t tests were performed to compare hypothermia (HT30) and normothermia (NT) tMCAO brain volumes of infarction (**e**, t(8) = 12.14, $P < 0.0001$), oedema (**f**, t(8) = 13.21, $P < 0.0001$), the neurological deficit scores (**g**, t(8) = 5.060, $P = 0.0010$), and fore-paw grip strength (**h**, t(8) = 5.608, $P = 0.0005$). Data were mean ± s.e.m ($n = 5$ mice). Source data are provided as a Source Data file.

Evoking hypothermia using chemogenetics 1 h after transient middle cerebral artery occlusion (tMCAO) significantly protected the mouse brain from ischaemia/reperfusion injury (Fig. 1d). The infarct core volume of normothermia tMCAO group (NT) compared with the hypothermia tMCAO group (HT30) was at $55.93 \pm 4.00\%$ vs. $6.97 \pm 0.51\%$ (Fig. 1e, $P < 0.0001$), achieving an almost 92% reduction in infarct volume due to hypothermia. Brain oedema of the hypothermic tMCAO group (HT30) was also significantly reduced compared with the normothermic tMCAO mice (NT) (Fig. 1f, $P < 0.0001$). Hypothermia significantly protected neurones in the cerebral cortex, as shown using crystal violet and Fluoro-Jade B (FJB) histochemical staining (Supplementary Fig. 1a–c). The neurological deficit score ($P = 0.0010$) and the forepaw pulling strength ($P = 0.0005$) of the HT30 group were also significantly better compared with the NT mice (Fig. 1g, h).

The hypothermic effects induced using chemogenetics were compared with those produced by surface cooling during post-tMCAO. We used physical surface cooling to lower $T_{core}$ to 33 °C and 29 °C, representing mild (>32 °C) and moderate (28–32 °C) hypothermia, respectively[10]. Mice were subjected to 0.5 h or 1 h tMCAO. We showed that both levels of hypothermia conferred significant brain protection (Supplementary Fig. 2a–g). Two-way ANOVA analyses showed that the level of $T_{core}$ reduction and tMCAO time significantly affected the tMCAO outcomes in tissue damage (Supplementary Fig. 2d, e; infarction volume: main interaction $F_{(2, 24)} = 13.02$, $P = 0.0001$; brain oedema volume: main interaction $F_{(2, 24)} = 27.36$, $P < 0.0001$). The motor functions were significantly affected by the level of $T_{core}$ reduction (Neurological deficit score: $F_{(2, 24)} = 20.12$, $P < 0.0001$; Forepaw grip strength: $F_{(2, 24)} = 49.13$, $P < 0.0001$), but not by the tMCAO time (Neurological deficit score: $F_{(1, 24)} = 5.765$, $P = 0.0245$; Forepaw grip strength: $F_{(1, 24)} = 10.88$, $P = 0.0030$). Neuroprotection was further demonstrated using brain sections stained with crystal violet and FJB (Supplementary Fig. 1a–c). Collectively, these data demonstrated that evoking hypothermia post-ischaemia potently protected the brain.

## Differences between hypothermia evoked by chemogenetics and surface cooling

Remarkable differences in physiological and metabolic responses occurred during hypothermia evoked by chemogenetics and surface cooling (Fig. 2; Supplementary Fig. 2h showing experimental designs). First, the brown adipose tissue temperature ($T_{BAT}$) dropped together with the $T_{core}$ after CNO injection (Fig. 2a), while $T_{BAT}$ remained higher and dropped slower than $T_{core}$ during surface cooling (Fig. 2b). Second, mice during chemogenetic-evoked hypothermia did not shiver, but surface-cooled mice shivered uncontrollably (Fig. 2c). The degree of shivering (numbers of tremors) inversely correlated with the drop of $T_{core}$ (Fig. 2c; Pearson $r = -0.9972$ for the C57BL/6J surface cooling mice; Supplementary Movie 1). Third, the noradrenaline level (Fig. 2d) and the blood glucose levels (Fig. 2e) increased during surface cooling-evoked hypothermia. Fourth, the respiration rate increased significantly as the $T_{core}$ was reduced further during surface cooling (Fig. 2f). In contrast, these physiological parameters remained little changed during chemogenetic-evoked hypothermia in Vglut2-Cre mice (Fig. 2c–f, $P > 0.05$).

We then determined the differences in metabolic rates during hypothermia using indirect calorimetry (Fig. 2g–i; Supplementary Fig. 2h, i). Chemogenetic-evoked hypothermia in mice showed the reduction of the volume of oxygen ($VO_2$) consumption by 59%, heat production by 60%, and the respiratory exchange ratio (RER) by 11% (Supplementary Fig. 2i–k). In contrast, the metabolic rate remained high during surface cooling-induced hypothermia, with $VO_2$ consumption reduced by 10%, heat production by 8%, and the RER level by 5% compared to normothermia (Fig. 2g–i, Supplementary Fig. 2i–k). The respiration rate (breaths per min) of mice in surface cooling-induced hypothermia was much higher than the chemogenetics-

evoked hypothermic mice (Fig. 2f). Together, these data demonstrated that WSNs-evoked hypothermia elicited a torpor-like state with reduced physiological and metabolic rates. In contrast, surface cooling was stressful and elicited strong shivering thermogenesis. These data indicated that stimulating WSNs may achieve tolerable therapeutic hypothermia in humans.

## Cerebral ischaemia disrupts $T_{core}$ circadian rhythm

To exclude the possibility that tMCAO itself causes therapeutic levels of hypothermia, we simultaneously recorded the core temperatures for the brain ($T_{cortex}$) and body ($T_{core}$) in free-moving mice (Supplementary Fig. 3a, b). The night-time $T_{cortex}$ was significantly lower than the $T_{core}$ ($P = 0.0015$, Supplementary Fig. 3c). The day-time $T_{cortex}$ was not different from the $T_{core}$ ($P = 0.1133$, Supplementary Fig. 3d). Pearson's correlation coefficient analyses also indicated a higher degree of similarity ($R^2$) between $T_{cortex}$ and $T_{core}$ during the day-time (Supplementary Fig. 3e, f). Therefore, all experiments were performed during the day-time when $T_{core}$ was the same as $T_{cortex}$.

We then measured the changes in core body temperature ($\Delta T_{core}$) after tMCAO. The mouse tMCAO model produced brain damage in the cerebral cortex and striatum but not in the POA[35]. The $\Delta T_{core}$ after 0.5 h or 1 h tMCAO (tMCAO-0.5 h group and tMCAO-1h group, respectively) was tracked during the 5 d reperfusion period (Supplementary Fig. 4a, b). Compared with the sham-operated mice, tMCAO at 1 d reperfusion exhibited a severe disruption of $T_{core}$ circadian rhythm (Supplementary Fig. 4c, f, g and Supplementary Table 1). The $\Delta T_{core}$ of tMCAO mice during the 1st day of reperfusion fluctuated ±1 °C from the mean $T_{core}$ of the normothermic sham mice (Supplementary Fig. 4i). The night-time $T_{core}$ recovered better than the day-time $T_{core}$ at 3 d and 5 d reperfusion (Supplementary Fig. 4d, e, i, and Supplementary Fig. 5). Eventually, $T_{core}$ recovered to the normothermic state after 5 d reperfusion (Supplementary Fig. 4h). These data demonstrated that tMCAO disrupted the $T_{core}$ circadian rhythm and evoked a spontaneous drop of $T_{core}$, albeit at a level not sufficient to achieve therapeutic hypothermia.

## DBS of MPN evokes deep hypothermia

DBS modulates neural circuits and has been used to treat multiple neurological disorders[36–41]. However, using high-frequency stimulation (HFS) to lower the $T_{core}$ for brain protection has not been attempted before[39,40]. Here, we used an open-loop electrode stimulation system implanted into the bilateral MPN (Bregma coordiantes ML/AP/DV: ± 0.3/+0.15/−5.15 mm) for DBS while tracking $T_{core}$ with an abdomen implanted telemetric temperature sensor or a rectal sensor (Fig. 3a–c, b' and Supplementary Fig. 6a–h). Stimulation settings were chosen empirically to maximise benefit. The HFS voltages were set at 1, 2, 3, and 4 V with 90 μs of pulse width at either 40, 100, or 130 Hz frequencies. HFS voltage (Pearson $r = -0.8391$) and frequency ($r = -0.7657$) dependently correlated with the reduction of $T_{core}$ to 32–34 °C within 30 min (Fig. 3d–g). The subsequent experiments were conducted with 4 V, 100 Hz at 90 μs of the pulse width. After switching off DBS, the $T_{core}$ gradually returned to normothermia within 2 h (Fig. 3h).

DBS mice at 33–34 °C exhibited reduced $VO_2$ consumption ($28\% \pm 0.08\%$), less heat production ($23\% \pm 0.05\%$), and reduced RER ($16\% \pm 0.04\%$) compared with the untreated control mice (Fig. 3i–l). The blood levels of glucose and norepinephrine did not change, and no signs of shivering occurred after DBS-induced hypothermia (Supplementary Fig. 7f–h). The respiration rate was significantly reduced during DBS compared with the surface cooling hypothermic mice (HT29 in Fig. 2f; $P < 0.0001$). The respiration rate of DBS mice was at a similar level to the chemogenetic-evoked hypothermic mice (HT30 + CNO, $P = 0.0327$) (cf. Fig. 2f). These metabolic and physiological parameters were remarkably similar to hypothermia evoked by DBS and the chemogenetic activation of WSNs.

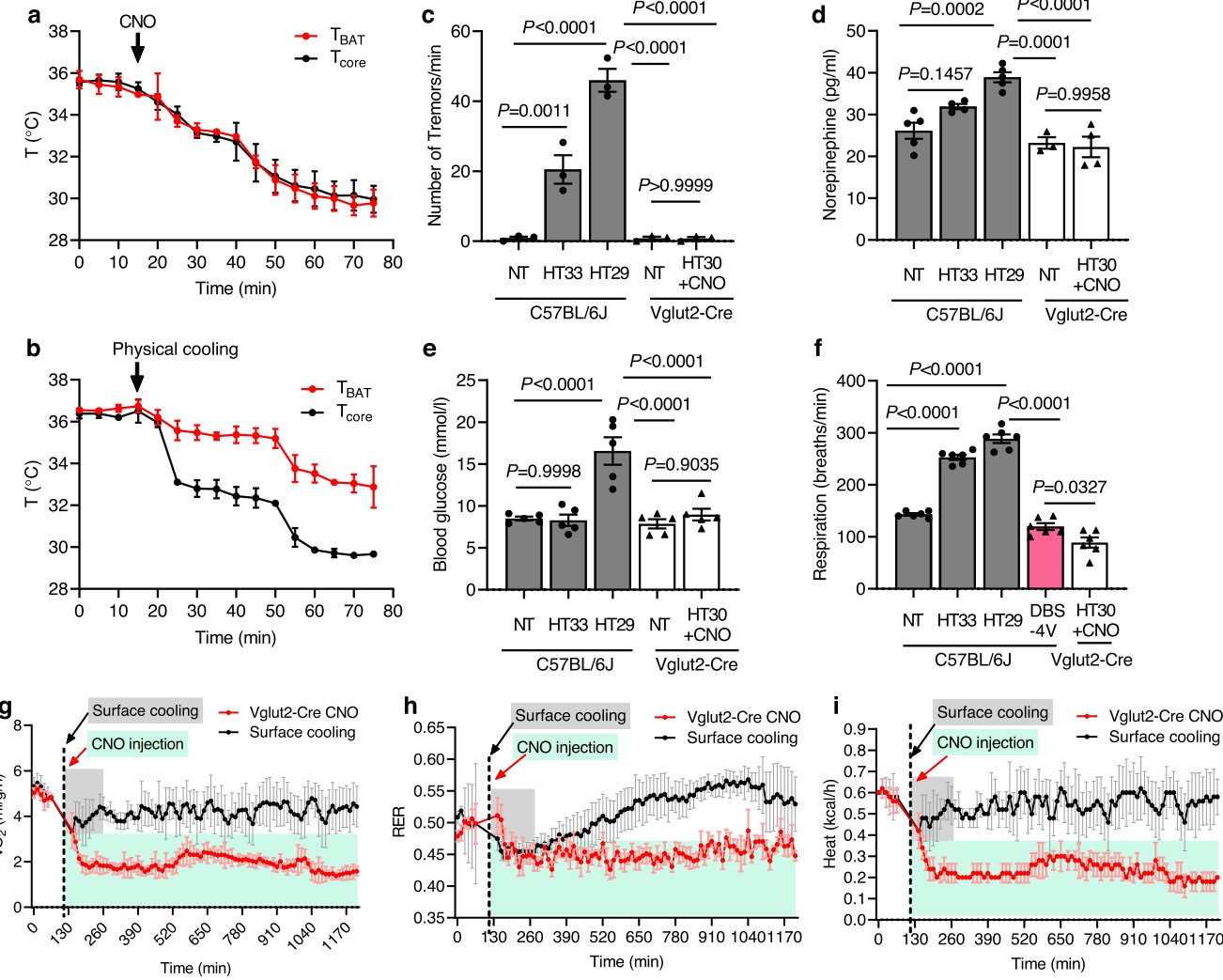

**Fig. 2 | Physiological and metabolic changes during hypothermia evoked by chemogenetics and surface cooling. a, b** The $T_{BAT}$ and corresponding $T_{core}$ during hypothermia were measured in mice ($n = 3$ mice). Hypothermia was produced in Vglut2-Cre mice injected with Gq-DREADD-AAV for 3 weeks and followed by CNO i.p. injection (**a**). Hypothermia was induced in C57BL/6J mice by surface cooling (**b**). **c** The numbers of tremors were recorded ($F_{(1, 12)} = 155.20$, $P < 0.0001$). **d** The blood levels of norepinephrine ($F_{(4, 25)} = 157.1$, $P < 0.0001$) and (**e**) glucose ($F_{(4, 20)} = 16.97$,

$P < 0.0001$) were measured. **f** The respiration rate was calculated ($F_{(4, 25)} = 157.1$, $P < 0.0001$). Statistical analyses used for **c–f** were 1ANOVA with Tukey's *post hoc* test for the C57BL/6J group and an unpaired two-tailed t test for the Vglu2-Cre-CNO mice. $VO_2$ consumption (**g**), RER (**h**), and heat level (**i**) were measured using indirect calorimetry ($n = 5$ mice). The quantifications of these measurements are shown in Supplementary Fig. 2i–k. All data were mean ± s.e.m. ($n = 5$ mice). Source data are provided as a Source Data file.

To further demonstrate the effectiveness of DBS, Vglut2-Cre mice pre-injected with AAV expressing Cre-dependent Gq-DREADD fused to mCherry were subjected to MPN DBS to induce hypothermia (34 °C) followed by the return to normothermia. These mice were then injected with CNO i.p. to induce deep hypothermia (32 °C) again (Supplementary Fig. 7i, j). These data demonstrated that DBS stimulation did not damage the MPN and that the preoptic WSNs were still functional after DBS.

### DBS of MPN evokes protective hypothermia
Importantly, DBS for 4 h in tMCAO mice at 1 h and 4 h after tMCAO conferred significant protections to the brain and preserved motor functions examined at 1 d and 3 d reperfusion (Fig. 3m–o). Compared with the tMCAO brains at 1 d and 3 d reperfusion, the DBS brain infarction volumes (Fig. 3n: $F_{(4, 20)} = 136.8$, $P < 0.0001$) and oedema (Fig. 3o: $F_{(4, 20)} = 91.98$, $P < 0.0001$) were significantly reduced. The neurological deficit scores (Fig. 3p: $F_{(4, 20)} = 8.636$, $P = 0.0003$) and the forepaw grip strength (Fig. 3q: $F_{(4, 20)} = 35.24$, $P < 0.0001$) were also better preserved. These data demonstrated that even delayed hypothermia using DBS was effective in protecting the ischaemic brain.

### DBS-mediated brain protection requires $T_{core}$ reduction
DBS of MPN led to a reduction in both $T_{core}$ and metabolism. To demonstrate that the reduction of $T_{core}$ is key to DBS-mediated brain protection, we performed 4 h DBS on tMCAO mice (1 h ischaemia) while maintaining the $T_{core}$ at 36.5 ± 0.5 °C throughout the DBS and reperfusion period (Fig. 4a–f). There were no differences between the tMCAO group (labelled as NT in Fig. 4b) and the DBS tMCAO group (labelled as DBS@36.5 °C in Fig. 4b) in terms of the infarct volume, neurological deficit scores, and forepaw pulling strength (Fig. 4c, e, f). The oedema volume was only marginally reduced in the DBS tMCAO group (Fig. 4d). These data showed that the reduction in the $T_{core}$ was key in DBS-mediated brain protection.

### DBS activates warm sensitive neurones
The mechanisms of DBS evoked hypothermia are complex. Here, we provide strong evidence to demonstrate that activation of WSNs contributed to DBS-triggered hypothermia.

First, since WSNs were primarily located in the medial preoptic area (MPA)[42], DBS of MPN in the MPA would potentially activate WSNs. To see if DBS in the neighbouring brain areas would also trigger

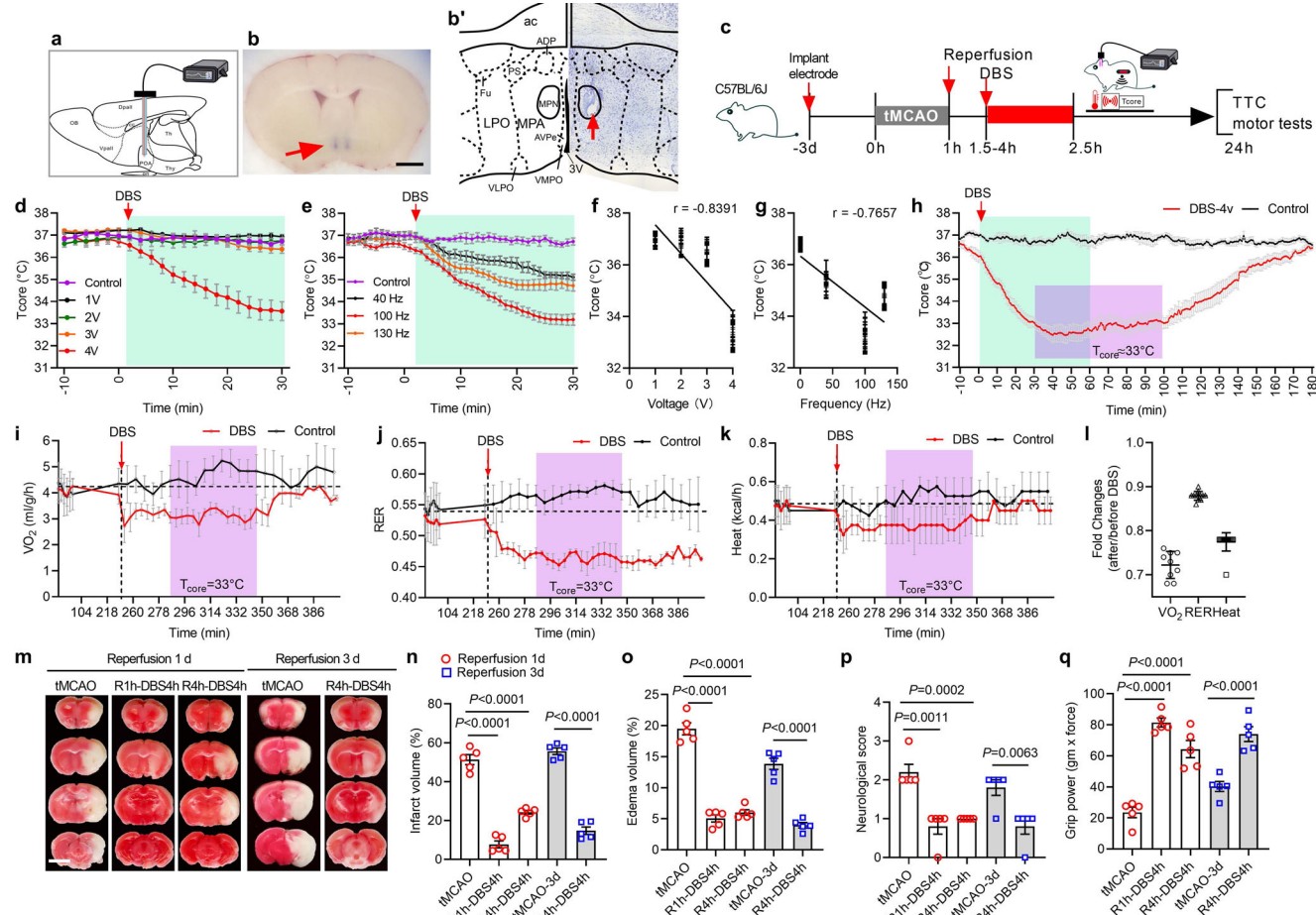

**Fig. 3 | Deep brain stimulation (DBS)-evoked hypothermia protected ischaemic brain. a** Schematic representation of a bipolar electrode trajectory to the bilateral medial preoptic nucleus (MPN). **b** Coronal brain section showing the preoptic area (POA) sites injected with Evens blue (blue dots, red arrow). Scale bar = 2 mm. **b′** Electrical lesioning for DBS stimulation site. The brain section stained with crystal violet was superimposed with outlines for the POA structures to indicate the location of MPN (red arrow). **c** Schematic of experimental design. **d** The plot showed the $T_{core}$ of DBS mice stimulated with various voltages at 100 Hz. **e** The plot showed $T_{core}$ of DBS mice stimulated with various frequencies at 4 V. The Pearson's correlation between the $T_{core}$ with voltages (**f**) and frequencies (**g**) was shown. **h** The entire time course of DBS-evoked hypothermia. The green-coloured areas in

**d**, **e**, **h** indicated the time-domain of DBS. The purple-coloured areas in **h**–**k** indicated the duration of stable $T_{core}$ at 33 °C. VO$_2$ consumption (**i**), RER (**j**), and heat level (**k**) of DBS mice (red coloured lines) and untreated control mice (black-coloured lines) were measured using indirect calorimetry (*n* = 5 mice). The quantifications of fold reduction during the targeted hypothermia time, as indicated by the purple-coloured areas, are shown in **l**. **m** Micrographs of coronal brain sections stained with TTC to show viable tissues (red colour) and damaged tissues (white colour). Scale bar = 4.5 mm. Quantifications of infarction volume (**n**), oedema volume (**o**), neurological deficit scores (**p**), and the forepaw grip strength (**q**). 1ANOVA with Tukey's *post hoc* test with specific *P* values as indicated in the graph. Data were mean ± s.e.m. (*n* = 5 mice). Source data are provided as a Source Data file.

hypothermia, electrodes were stereotaxically inserted into the lateral preoptic nucleus (LPO: ML/AP/DV:±0.8/+0.15/−5.15 mm), the ventral pallidum (VP:±1.2/+0.15/−5.15 mm), and an aera not known to contain WSNs – the vertical limb of the diagonal band of Broca (VDB: ML/AP/DV:±0.3/+0.74/−5.15 mm) (Fig. 4g, j). DBS stimulation of neither LPO, VP, nor VDB produced a therapeutic level of hypothermia below 34 °C (Fig. 4h, i, k). Only shallow reduction in $T_{core}$ occurred during stimulating the LPO and VDB, suggesting that MPN is a specific DBS targeting area to evoke hypothermia.

Second, DBS elicited the co-expressions of c-Fos proteins with specific biomarkers for WSNs, including the brain-derived neurotrophic factor (BDNF) and *Adcyap1*[18,21,27,43]. DBS-induced the expression of c-Fos was mostly in the MPA (81%) and MPN (14.9%) of the POA (Fig. 5a–d, h) and which was significantly reduced in the adjacent LPO, median preoptic nucleus (MnPO) and ventromedial preoptic nucleus (VMPO) (Fig. 5a, c, h). Most c-Fos positive cells in the MPN showed co-localisation with BDNF (81.9%; Fig. 5f, i), and 73.6% of BDNF cells were also co-localised with c-Fos. The *Adcyap1* is a specific marker of a subset of glutamatergic WSNs[21], and its expression mainly occurs in the bilateral MPAs, as indicated by in situ hybridisation using RNAscope

probes (Fig. 5b, d). Only 22.4% of *c-Fos* mRNA positive cells in MPN showed co-localisation with the *Adcyap1* mRNA, and 21.9% of *Adcyap1* mRNA positive cells showed the co-localisation with *c-Fos* mRNA expression (Fig. 5g, j). These data demonstrated that DBS activated WSNs in the MPA, which contributed to the induction of hypothermia.

Third, to unequivocally demonstrate that DBS activation of WSNs is critical in inducing therapeutic level of hypothermia, inhibitory AAV2/9-hSyn-DIO-hM4D(Gi)-eGFP-WPRE-pA was injected into the MPN of Adcyap1-2A-Cre knock-in (B6.Cg-*Adcyap1*$^{tm1.1(cre)Hze}$/ZakJ) and Vglut2-ires-cre knock-in (C57BL/6J) mice to silence Adcyap1 and Vglut2 neurons, respectively. The injected inhibitory AAVs were primarily expressed in the bilateral POA (Fig. 4m, n). Silencing of excitatory Vglut2 neurones completely blocked DBS-evoked hypothermia (Fig. 4o). Interestingly, inhibiting the *Adcyap1* + neurones in the MPN partially blocked DBS-evoked hypothermia (Fig. 4p), suggesting the involvement of other WSNs in the MPA in response to DBS-stimulation.

Fourth, DBS directly activated WSNs in vitro. We sampled neurones electro-physiologically in the MPN of C57BL/6J mice brain slice to show a subpopulation of neurones with increased AP firing rate in response to elevating voltage HFS from 0.5 to 8 V (Fig. 6a, b)[42]. A

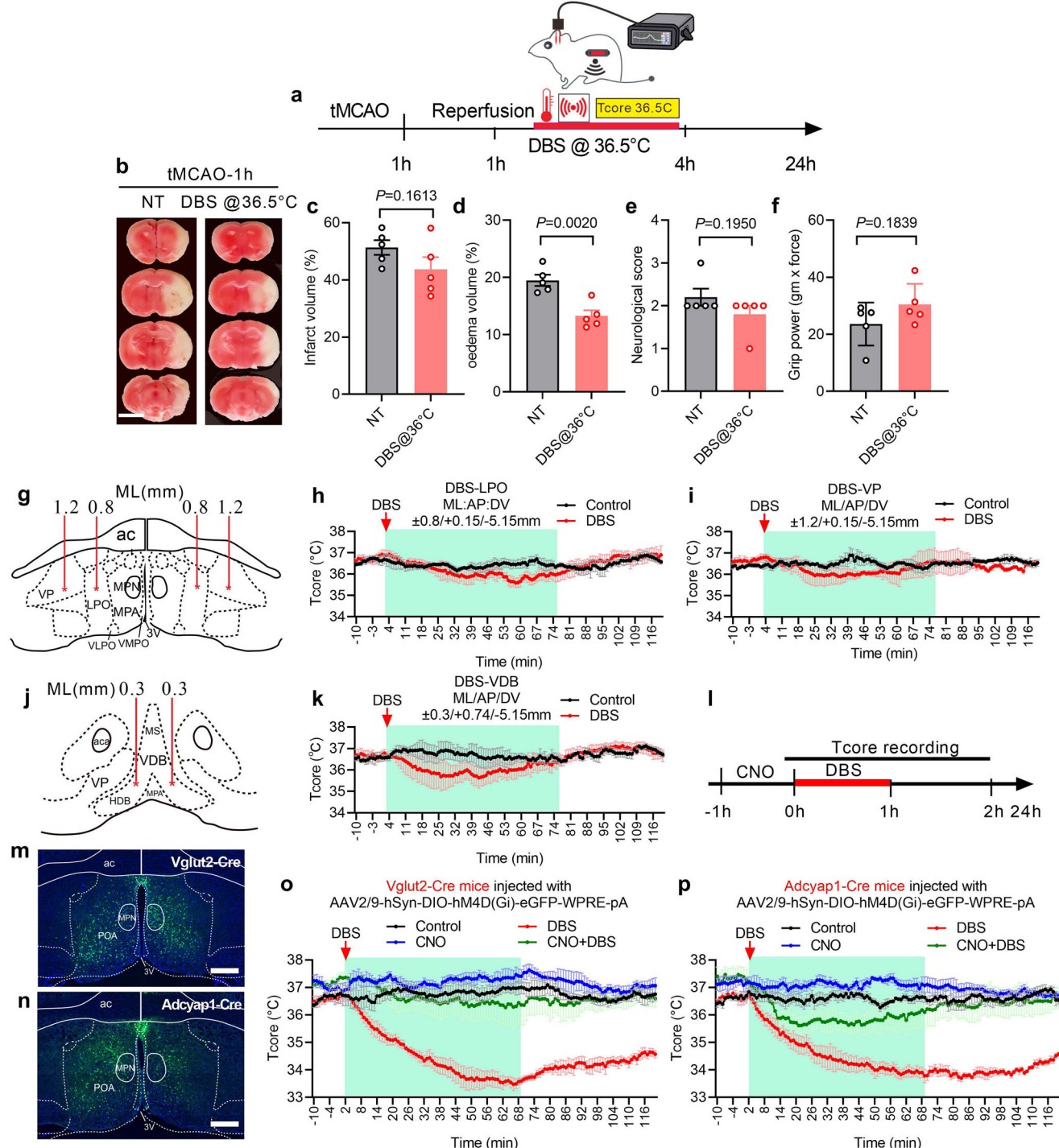

**Fig. 4 | Determination of DBS selectivity and specificity in $T_{core}$ reduction for brain protection. a** Schematic representation of DBS on MPN with $T_{core}$ maintained at 36.5 °C. **b** Brain coronal sections at 2 mm thickness were stained with TTC to show viable tissue (red colour) and damaged tissue (white colour). Scale bar = 5 mm. **c–f** Unpaired two-tailed t tests were performed to compare normothermic tMCAO (NT) and DBS of tMCAO brain at 36.5 °C (DBS@36.5 °C) with volumes of infarction (**c**, t(8) = 1.543, P = 0.1613), oedema (**d**, t(8) = 4.495, P = 0.002), the neurological deficit scores (**e**, t(8) = 1.414, P = 0.1950), and forepaw grip strength (**f**, t(8) = 1.455, P = 0.1839). Data were mean ± s.e.m (n = 5 mice). **g**,**j** Schematic of the POA nucleus on coronal brain sections and the locations of the bilateral DBS

electrodes. **h**, **i**, **k** The plots show the $T_{core}$ of DBS mice stimulated with 4 V 100 Hz at the LPO (**h**), VP (**i**), and VDB (**k**). ML mediolateral, AP anteroposterior, DV dorsoventral, ac anterior commissure. **l** Schematic of $T_{core}$ recording with CNO and DBS treatment. **m** Coronal brain section with outlines of sub-anatomical structures showing injected inhibitory AAVs (EGFP) expressed in the bilateral POAs of Vglut2-Cre mice and Adcyap1-Cre mice brain (**n**). The blue colour came from DAPI staining. Scale bars = 100 μm. **o** The plot of $T_{core}$ of Vglut2-Cre mice injected with an inhibitory AAV. **p** The plot of $T_{core}$ of Adcyap1-Cre mice injected with an inhibitory AAV. Data were mean ± s.e.m (n = 4 mice). Source data are provided as a Source Data file.

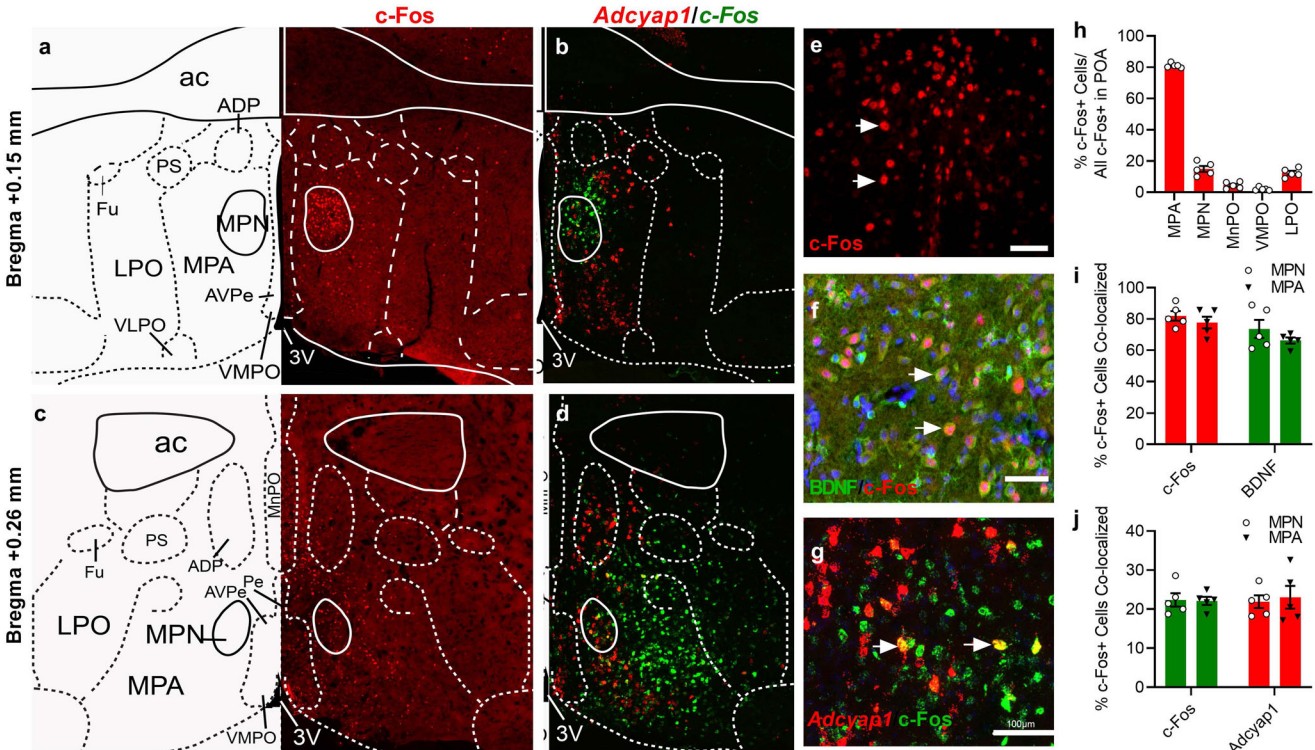

**Fig. 5 | DBS-induced expression of WSN markers. a, c** Immunohistochemical staining of c-Fos on coronal brain sections at the bregma point as indicated. **b, d** In situ RNAScope labelling of *Adcyap1* and *c-Fos* mRNAs. **e–g** Higher magnification images of c-Fos staining (**c**), and the co-localisation of c-Fos with BDNF (**f**) and *c-Fos* with *Adcyap1*. **h** Quantifications of the ratio of c-Fos positive cells in the indicated area against the total numbers in the POA. **i** The ratio of co-localisation between c-Fos and BDNF in the MPN and MPA. **j** The ratio of co-localisation between *c-Fos* and *Adcyap1* in the MPN and MPA. Scale bar = 100 μm. Data were mean ± s.e.m (*n* = 5 mice). Source data are provided as a Source Data file.

synaptic transmission inhibitor cocktail, including 50 μM D-2-amino-5-phosphonovaleric acid (D-AP-5), 10 μM 6-cyano-7-nitroquinoxaline-2,3-dione (CNQX), and bicuculline (20 μM), completely ameliorated AP firing evoked by HFS (Fig. 6c, d). The specific N-type voltage-gated calcium channel (N-VGCC) inhibitor, ω-conotoxin-GVIA (2.5 μM), also blocked 4 V HFS evoked AP (Fig. 6d, the right panel). Furthermore, electrophysiological recording of mCherry-positive neurones from slices of Vglut2-Cre mice brain injected with Gq-DREADD-AAV virus showed that 83% of the neurones had increased voltage-dependent AP firing rate (Supplementary Fig. 8a–c, Pearson r = 0.915), indicating HFS activated excitatory WSNs.

We then determined the responses of POA neurones on the brain slices to increasing temperature to identify WSNs, moderately warm-sensitive neurones (MSNs), temperature-insensitive neurones (ISNs), and silent neurones firing no APs (silent cell) (Fig. 6e–g). The defining feature of WSNs is the thermal coefficient (TC), expressed as the firing rate per the function of temperature (impulses s$^{-1}$ °C$^{-1}$). A minimum TC of at least +0.8 impulses s$^{-1}$ °C$^{-1}$ was applied to identify WSNs. Neurones with a TC in between +0.2 and +0.8 were regarded as MSNs and neurones with TC below +0.2 were regarded as ISNs[28,44,45]. Neurones without spontaneous firing were silent cells. Increasing brain slice temperature from 26 °C to 33, 36, and 39 °C allowed the determination that 20% of cells were WSNs (Fig. 6f, g), 22.5% were MSNs, 40% were ISNs, and 17.5% were silent cells, confirming a previous report[28]. Silent cells did not respond to increasing temperature (Fig. 6f).

Next, we subjected the brain slices to a sequence of 4 V HFS followed by thermal stimulations from 33 to 39 °C. This sequence of stimulations allowed us to determine WSNs which were responsive to both electrical and thermal stimulations with a significant increase in firing rates [Fig. 6h; RM-1ANOVA and Dunnett's *post hoc* test F$_{(13, 39)}$ = 5.877, 4 V HFS (*P* < 0.05) and 39 °C (*P* < 0.0001)]. The data showed that 27.5% of neurones responded

to both 4 V HFS and 39 °C thermal stimulation (Figs. 6i) and 23.6% of neurones only responded to 4 V HFS, but not thermal stimulation. About 15.7% of neurones were WSNs that did not respond to 4 V HFS. Collectively, these data demonstrated that 4 V HFS indeed activated WSNs in the MPN.

Fifth, synaptic vesicle exocytosis is another indication of neural activity in response to HFS. We used FM1-43 to label presynaptic readily releasable vesicles on brain slices. Stimulation with 4 V HFS significantly increased excitatory synaptic vesicle release compared with the non-stimulated and 1 V HFS groups (Supplementary Fig. 8d–f). The N-VGCC inhibitor, ω-conotoxin-GVIA (2.5 μM), also inhibited the 4 V HFS-elicited synaptic vesicle release (Supplementary Fig. 8e, f). Two-way ANOVA revealed that there was a statistically significant interaction between the effects of 4 V HFS and ω-conotoxin-GVIA inhibition (F$_{(3, 27)}$ = 12.08, *P* < 0.0001). Both 4 V and ω-conotoxin-GVIA had a statistically significant main effect on synaptic vesicle exocytosis (*P*$_{(No stim vs. 4V)}$ < 0.0001; *P*$_{(4V vs. GVIA-4V)}$ = 0.0002, respectively).

Sixth, neuronal intracellular calcium transients in response to HFS and thermal stimulations were measured using Fluo-8 on acute brain slices from C57BL/6J and Vglut2-Cre mice injected with the Gq-DREADD-AAV. We found that HFS significantly increased the average intensity (dF/F0) and frequency (min) of intracellular calcium transients ([Ca$^{2+}$]i) in a subpopulation of neurones in the MPN (Fig. 6j, k; Supplementary Fig. 9c–e). The cocktail inhibitors to both the excitatory and inhibitory synapses, including D-AP-5 (50 μM), CNQX (10 μM), and bicuculline (20 μM), blocked [Ca$^{2+}$]i in response to 4 V HFS (Fig. 6i–l; Supplementary Fig. 9g). Blocking the N-VGCC using ω-conotoxin-GVIA also inhibited [Ca$^{2+}$]i (Fig. 6k, l). A 39 °C thermal stimulus re-activated 53% of the 4 V HFS responsive cells, suggesting that these were WSNs responsive to both stimuli (Supplementary Fig. 9a, b, f).

Finally, to exclude the possibility that stimulating electrodes might produce heat high enough to activate warm sensitive neurones

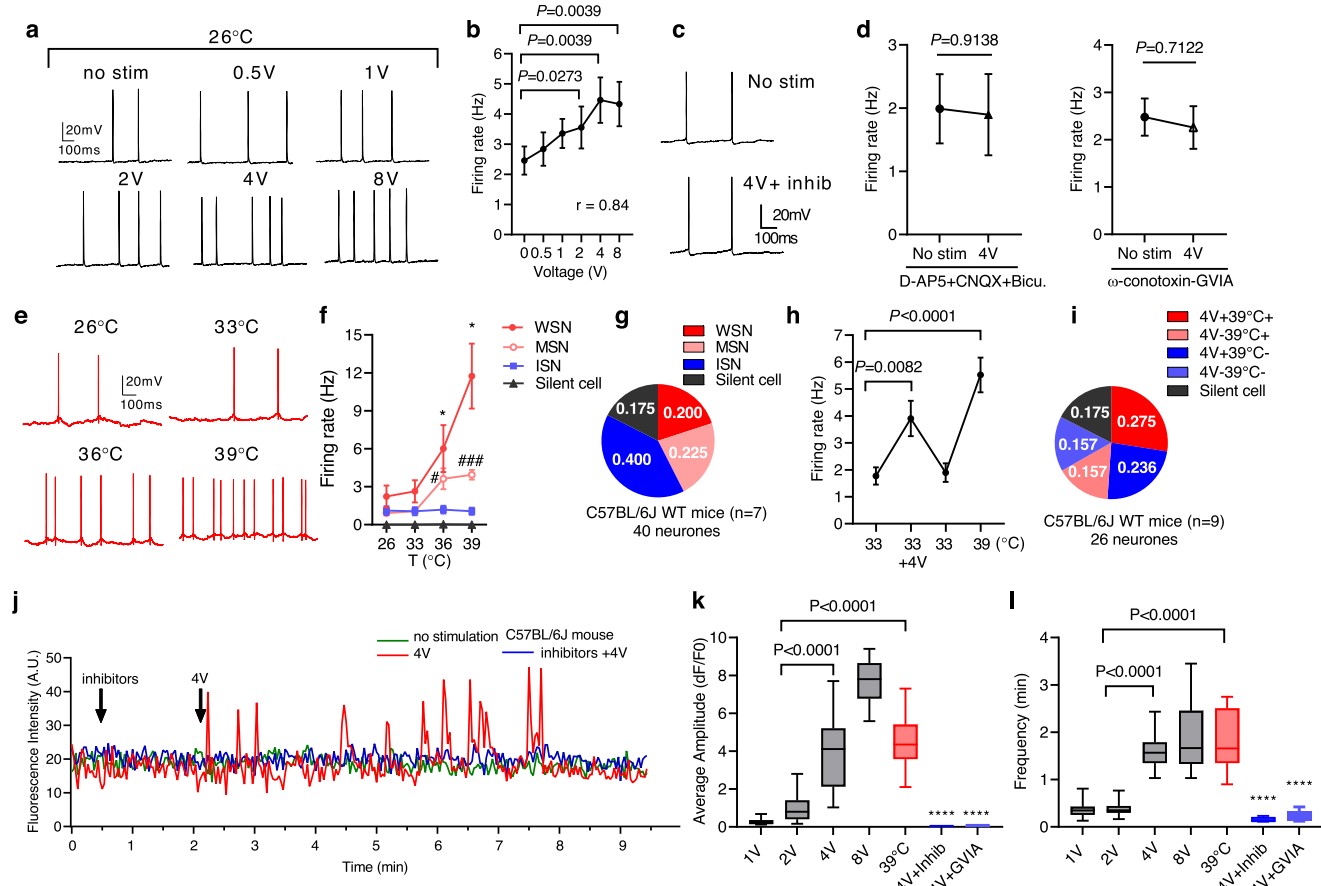

Fig. 6 | DBS activated warm sensitive neurones (WSNs) in the MPN. a Traces of action potential (AP) firing induced by HFS of the POA neurones. b AP firing rates in the POA of C57BL/6J mice brain slices (two-tailed Wilcoxon test, $n = 9$ cells from 4 mice). c, d A cocktail of synaptic blockers containing D-AP-5, CNQX, and biculline (c and d left panel: unpaired two-tailed t test, t(14) = 0.1102, $P > 0.05$, $n = 8$ cells from 3 mice), and ω-conotoxin-GVIA (d right panel: t(40) = 0.3716, $P > 0.05$, $n = 24$ cells from 4 mice) eliminated AP firing in response to 4 V HFS. e Traces of AP firing evoked by thermal stimulations of the MPN neurones. f MPN neuronal type based on AP firing rate and TC. RM-2ANOVA with Tukey's multiple comparisons test was performed with * indicating WSNs $P_{(39 °C \text{ vs. } 26 °C)} = 0.0132$ and $P_{(36 °C \text{ vs. } 26 °C)} = 0.039_6$, # indicating MSNs $P_{(36 °C \text{ vs. } 26 °C)} = 0.0489$, and ### for $P_{(39 °C \text{ vs. } 26 °C)} = 0.0003$. $n = 40$ cells from 7 mice. g POA cell categories based on their TC. h Neuronal AP firing rate evoked by a sequence of thermal and electrical

stimulations (RM-1ANOVA with Dunnett's post hoc test, $F_{(1.528, 19.86)} = 26.56$, $P < 0.0001$; $n = 26$ cells from 9 mice). i The proportions of POA neurones responded to 4 V HFS and/or thermal stimulation. j Wild-type mouse brain slices calcium transients without HFS (green line), with 4 V HFS (red line), and with a cocktail of synaptic blockers (black line). Quantification of the average calcium transient amplitude (dF/F0) (k) and calcium transient spike frequency (min) (l) in response to HFS and in the presence of the cocktail of synaptic inhibitors (inhib) or ω-conotoxin (GVIA). Box plots indicate median (middle line), 25th, 75th percentile (box) and the maximum and minimum values as the whiskers. 1ANOVA with Tukey's post hoc test; ****$P < 0.0001$ compared with the 4 V group for (k, l). $n = 20$ cells from 5 mice. All data were mean ± s.e.m. Source data are provided as a Source Data file.

at the stimulation site, we implanted the stimulating electrode and a fine thermocouple together into the brain (Supplementary Fig. 7a). The recorded local brain temperature did not change during 2–12 V HFS DBS (Supplementary Fig. 7b–e). Furthermore, cessation of DBS-evoked hypothermia in Vglut2-Cre mice injected with the Gq-DREADD-AAV could be re-induced by injecting CNO to lower the $T_{core}$ to 30 °C (Supplementary Fig. 7i, j), indicating DBS did not damage the MPN while driving body cooling.

Together, the in vivo DBS and in vitro brain slice experiments demonstrated that HFS of the MPN activated warm sensitive neurones, which contributed to the induction of hypothermia and brain protection against cerebral ischaemia. The axonal and synaptic mechanisms may also contribute to DBS-evoked hypothermia.

## Discussion

The advantage of DBS-induced hypothermia is that DBS exhibits rapid onset, voltage dependence, and significant therapeutic benefits. Even delayed DBS after cerebral ischaemia was also beneficial for brain protection. The lack of shivering thermogenesis means that

stroke patients could potentially tolerate well hypothermia evoked by DBS. Therefore, further refinement of this technique is warranted for clinical use. We chose DBS stimulation and avoided using direct thermal stimulation of the MPN. Direct thermal stimulation of the POA can modify the temperature over a wide range of hypothalamic sites outside the POA[46], creating potential side effects. The pharmacological approach to therapeutic hypothermia is also not as specific as DBS. For example, pharmacological activation of the A1 adenosine receptor has been shown to induce hypothermia in rats exposed to cold ambient temperatures[47]. Sedative drugs, such as barbiturates, are also known to cause hypothermia[47]. However, these agents have unintended targets and side effects. Barbiturate-mediated neuroprotection is through many mechanisms, including the redistribution of cerebral blood flow, sodium channel and glutamate receptor blockade, calcium influx, free radical formation, and potentiation of GABAergic activity, not to mention on the reduction of metabolism[48,49]. In contrast, DBS-evoked hypothermia is specific to the preoptic WSNs, and the level of hypothermia is controllable throughout the process.

It remains unclear how DBS hypothermia suppresses shivering. Classic works by Hemingway et al. (1954)[50] and Anderson et al. (1956)[51] using anaesthetised cats and non-anaesthetised goats, respectively, showed the existence of a possible centre for shivering inhibition, or "heat loss centre" in the preoptic region of the hypothalamus. Electrical stimulation of specific points within the hypothalamus inhibits shivering without the complication of the musculature, which is needed for skeletal movement[50]. Repeated stimulation of this "heat loss centre" lowered the body temperature[51]. Interestingly, stimulation of the "heat loss centre" did not affect blood pressure, heart rate, or blood sugar concentration, and behavioural changes were also slight or absent. These early pioneering works lend further support to our observations that DBS of POA rendered mice hypothermia without signs of shivering and the appearance of many adverse physiological effects.

The present study also provided strong evidence demonstrating that activation of WSNs is directly responsible for DBS-evoked hypothermia. First, DBS activated the co-localised expressions of c-Fos proteins with markers for WSNs in the POA, such as BDNF proteins and *Adcyap1*mRNAs. The c-Fos expression was primarily restricted in the medial preoptic area and not in the adjacent LPO, where the expression of WSNs markers is also reduced. This data indicates that DBS elicited a localised response in the POA. Second, chemogenetic inhibition of Vglut2 expression on excitatory neurons in the POA abolished DBS-evoked hypothermia. Interestingly, inhibiting Adcyap1 expressing neurones did not completely ameliorate DBS-evoked hypothermia (Fig. 4m). The resulting hypothermia was shallow and not as deep as by inhibiting Vglut2+ neurones, indicating other types of WSNs might also be involved. Third, the DBS electrode was placed mediolaterally from the MPN into the LPO, VP, and anteroposterior to the VDB. As shown in Fig. 4g–j, stimulating these areas did not induce deep hypothermia. Nevertheless, it is interesting that DBS stimulation of LPO and VDB elicited a shallow reduction in $T_{core}$ not lower than 35 °C, supporting recent reports that LPO neurones also have an important role in regulating $T_{core}$ in addition to wake-sleep[25,52,53]. Fourth, we show that maintaining $T_{core}$ at normothermia during DBS abolished brain protection against tMCAO, which strongly suggests that DBS-induced temperature changes are essential for brain protection.

The in vitro evidence from brain slices also showed that DBS triggered rapid changes in membrane potential of excitatory WSNs with a temperature-dependent increase in AP firing rates and calcium transients, underlying a potential mechanism of DBS-evoked hypothermic response. Synaptic blockers eliminated WSNs' responses to electrical stimulation, indicating that the WSNs' modulatory networks are also crucial in DBS-evoked hypothermia[28]. The mechanism of DBS, in general, is complex and unclear but may include the classic rate model, the local adenosine release, and neuroprotection, as shown in Parkinson's disease[54–56]. Our data demonstrated that DBS also directly affects WSN membrane potential and its synaptic networks[28].

Stroke is a leading cause of mortality and disability worldwide[3]. Stroke therapeutics represents the most significant unmet medical needs[1] and emphasizes the urgent demand for new mechanistically validated targets. The present study established a proof of concept to use DBS-evoked moderate hypothermia for brain protection. Further studies are warranted to translate this technique into human stroke therapeutics.

## Methods
### Chemicals
Unless stated otherwise, all chemicals were purchased from Sigma–Aldrich (St Louis, MO, USA). TTC: 2,3,5-triphenyl tetrazolium chloride; BSA: bovine serum albumin; FJB: Fluoro-Jade B. Also see the Supplementary Source file for the listed Key Reagents.

### Animals
C57BL/6J mice were purchased from the Vital River Laboratory Animal Technology Co., Ltd. (Zhejiang, China). Vglut2-Cre knock-in mice (strain #028863, B6J.129S6(FVB)-*Slc17a6*^tm2(cre)Lowl^/MwarJ) and Adcyap1-Cre mice (strain #030155, B6.Cg-*Adcyap1*^tm1.1(cre)Hze^/ZakJ) were purchased from the Jackson Laboratory (USA). Mice were maintained in a condition-controlled room in a pathogen-free SPFII animal facility (23 ± 1 °C, 50 ± 10% humidity). A 12 h light/dark cycle was automatically imposed. Mice were housed in groups of four to five per individually ventilated cages and given access to food and water *ad libitum*. Experimenters were blinded to animals' treatments and sample processing throughout the subsequent experimentation and analyses.

### Ethical approval and animal experimentation design
Animal experiment protocols were approved by the Animal Care Committee of the Southern University of Science and Technology (Shenzhen, China). The ARRIVE guideline was followed when designing, performing, and reporting animal experimentation[57]. Efforts were made to minimise the number of mice used. Mice used in the current study were randomly assigned to each group to maintain total randomisation. After the surgery, mice in each group were randomly assigned to receive hypothermia treatment or without hypothermia as a control. The inclusion criterion was based on the identical age and sex of the mice and that the stroke model must be successful based on the initial neurological test score of ≥2 at 30 min after the stroke surgery. Mice without successful stroke will be excluded from the study to minimise the possible random variability in stroke size and motor deficits. Mouse failed to survive tMCAO surgery at the end of the 14 d periods of the experimentation were also excluded. The DBS and chemogenetics experiments used both sexes of mice randomly assigned into the groups. However, the tMCAO experiments were performed using only male mice, as the female mice would produce inconsistent stroke infarctions depending on the blood level of oestrogen and hormonal cycle stages, which is well documented in the literature[35].

To achieve meaningful statistical differences, a minimum of five mice per group were used in the in vivo experiments, including temperature recordings, hypothermia experimentations, metabolic monitoring, DBS studies, in vitro brain slice electrophysiology recordings, and calcium imaging. For tissue section staining, at least three mice per group were used. C57BL/6J mice and Vglut2-Cre mice with the same age (3-month-old) and sex (male) were first randomly assigned to two groups of Sham-operated and tMCAO. After the surgery, mice in each group were randomly assigned to receive hypothermia evoked by CNO injection, surface cooling, and DBS treatment.

At the end of the experimentation, to obtain brain tissues for electrophysiological studies, animals were euthanized by cervical dislocation and brains were immediately harvested. Alternatively, to obtain brain tissues for sectioning, staining, and RNAScope analysis, mice were administered 1.5% sodium pentobarbital at a dose of 0.06 ml per 10 g body weight through intraperitoneal injection at the lower left or right quadra of the abdomen. After 5 min, mice were euthanized by transcardiac perfusion with 0.01 M PBS, and 4% freshly made paraformaldehyde under deep anaesthesia. Data derived from all qualified animals were included in the analyses and presentation of the results.

### Transient occlusion of the middle cerebral artery (tMCAO) model
Mice were anaesthetized with a mixture of 2.5% isoflurane in N2:O2 (70:30; flow rate 400 ml/min) in an induction chamber (Product #R540, RWD Life Science, Shenzhen, China). When the animal showed no sign of consciousness, it was moved to the anaesthetic face mask supplied with 1.5% isoflurane in N2:O2 (70:30; flow rate 400 ml/min). The animal's body temperature was maintained at

36.5 ± 0.5 °C using a rectal temperature probe and a heating blanket (Product #TCAT-2 Temperature Controller, Physitemp, USA). Furs around the ventral neck region were shaved using a hair clipper to expose the skin, which was disinfected with iodine followed by 75% ethanol. Under the operating microscope, tMCAO by intraluminal occlusion of the left middle cerebral artery (MCA) was performed as previously described using male mice at 8–10 weeks of age at a 25–30 g of body weight[58–60]. Briefly, an 8-0 monofilament (Jialing Biotechnology Company, Guangzhou, China) with a silicon-coated tip was inserted into the MCA. After 60 min MCAO, the filament was withdrawn, and the blood flow was allowed to resume to basal levels, as assessed by laser Doppler flowmetry. After suturing the skin, the mice were kept in an intensive care unit cage with controlled oxygen and temperature. In the sham group, mice were subjected to the same surgery without MCA occlusion.

## Neurological deficit scores and grip strength measurements
A six-point scale assessment of neurological deficits and forelimb grip strength test were performed as previously described[58–60]. The neurological deficit score was assessed by an individual blinded to the treatment of the mice 30 min after MCAO when animals were fully awake after anaesthesia as follows: 0, normal motor function; 1, flexion of the contralateral torso and forelimb on lifting the animal by the tail; 2, circling to the contralateral side but normal posture at rest; 3, leaning to the contralateral side at rest; 4, no spontaneous motor activity; and 5, death. The forelimb grip strength test was performed using the Grip Strength Meter from Columbus Instruments (BAS-47200, Ugo Basile, Italy), which measures forepaw muscle strength and neuromuscular integration relating to the grasping reflex in the forepaws. The peak preamplifier automatically stores the peak pull force and shows it on a liquid crystal display. For each animal, at least three measurements were taken at a specific time point, and the mean and standard error were calculated.

## Measurements of cerebral infarction and oedema volume
Infarct size was measured by a colorimetric staining method using 2,3,5-triphenyl tetrazolium chloride (TTC) as described previously[58–60]. Briefly, brains were dissected and cut into four 2 mm thick coronal slices, which were stained with 5 ml of 2% TTC for 10 min at 37 °C. Brain slices were imaged using a photo scanner (HP ScanjetG4010). The infarct size was measured using Image J based on the measurements of the volume of the white-coloured areas in the ipsilateral side of the brain x the length of the brain slices. The oedema volume was subtracted to derive at the final ipsilateral side of the infarction. The ratio of brain infarction was calculated as previously described[61]. Briefly, the following equation was used to determine the infarction ratio: Percentage of infarct = (volume of the contralateral hemisphere – (volume of intact ipsilateral hemisphere – the volume of ipsilateral infarct hemisphere)/volume of intact ipsilateral hemisphere) × 100; Percentage of oedema = (volume of the ipsilateral hemisphere – the volume of the contralateral hemisphere)/volume of contralateral hemisphere × 100.

## Rectal thermometry
Rectal thermometry is a precise and straightforward method of measuring short-term core body temperature in rodents. The mouse was hand-restrained, and a thermocouple probe (covered with Vaseline) was gently inserted into the rectum at 2 cm depth. The wire was fixed on the tail with medical water-proof fabric. Flexible plastic-covered thermocouple wires had good deformability allowing the mouse to move freely in the home cage while connected to the thermometer (DC700 multichannel temperature recorder, DCUU, Pumei, China) through the long wire. The mice receiving the rectal thermo-probe showed no discomfort or resistant behaviours while monitoring the rectal temperature.

## Telemetric measurement of core body temperature
The core body temperature of free-moving mice was tracked using a telemetry system (Data Sciences International, MN, USA) with compatible temperature transponders implanted into mice. Mice were anaesthetized with 1.5% isoflurane, and a sterile telemetric transmitter (TA-F10, DSI, USA) was implanted in the abdominal cavity. Afterward, muscle and skin layers were sutured separately with absorbable surgical threads. Subcutaneous injection of carprofen (5 mg/kg) alleviated the surgery pain of mice. After seven days of recovery, mice were used for experiments to determine body temperature via the telemetric recording system. Radio signals encoding core body temperature at 25 °C ambient temperature were obtained by receiver plates under the home cages (RSC-1, DSI, USA) using Ponemah Physiology Platform V6.41.20418.1 software (DSI, USA).

## Brain temperature measurement
The deep brain temperature was detected by the implanted thermocouple probes into the cortex. Mice were anaesthetized with 1.5% isoflurane and placed on a stereotaxic apparatus. The eyes were protected with moisturising ointment. After removing the fur on the head around the region of the interest, a burr hole was drilled on the exposed skull with a frame-mounted drill. An ultra-thin thermocouple (0.13 mm diameter and 0.23 mm sensor tip diameter; IT-24P; Physitemp, USA) was implanted in the cortex of the left brain (coordinates from bregma: ML: ± 0.5 mm, AP: + 0.8 mm, and −1.3 mm below the bregma). The thermo-probe was secured to the skull using dental cement together with two stainless-steel screws threaded into the skull. After recovery for 3 d, the mice were subjected to the MCAO surgery. The left side MCA was occluded for 0.5 h or 1 h. The blood flow was allowed to resume afterward for reperfusion up to 3 d. The core body temperature of mice was recorded by the rectal thermometry, and the brain core temperature was tracked by the implanted thermo-probe connected to the thermometer (DC700 multichannel temperature recorder) at room temperature 25 ± 0.5 °C.

## Determination of brown fat tissue temperature ($T_{BAT}$)
The interscapular BAT (iBAT) was exposed under aseptic surgery as described[62,63] and a sterile thermocouple probe was inserted in the iBAT of mice anaesthetised with 1.5% isoflurane. The thermocouple line was fixed with iBAT and the skin with 3 M™ Scotch-Weld™ Surface Insensitive Instant Adhesive SI Gel (3 M, USA) and sutured together with the skin. The $T_{BAT}$ of mice was recorded in real-time by connecting the thermos-probe to the thermometer (DC700 multichannel temperature recorder) at room temperature (25 ± 0.5 °C). Data were collected and analysed using GraphPad (version 9.0.0, USA).

## Surface cooling evoked hypothermia
We sprayed mice in their home cage with an implanted telemetric transmitter or rectal thermo-probe with cool water. A blowing fan aided the cooling process until reaching the core body temperature of 29° or 33 °C. The cooling process continued for the duration of the required experimentation.

## Chemogenetic hypothermia and stereotactic DREADD viral infection
To determine activation of WSNs in the MPN can trigger a torpor-like state, a chemogenetics approach was employed to selectively activate WSNs in the POA, followed by the assessment of various physiological and pharmacological characteristics. Briefly, Vglut2-ires-Cre mice were bilaterally injected with AAV particles (serotype 9, 120 nl) into the MPN (coordinates from bregma: ML: ± 0.3 mm, AP: + 0.15 mm, and DV: −5.15 mm) following the established method[27]. The AAV particles were the Gq-DREADD [AAV2/9-hSyn-DIO-hM3D(Gq)-EGFP (or mCherry) and vector control AAV2/9-hSyn-DIO-EGFP (or mCherry). All AAV viruses were purchased from Taitool Bioscience Co. Ltd (Shanghai, China).

AAVs were diluted with PBS to a final concentration between $5 \times 10^{12}$ and $1 \times 10^{13}$ genome copies per ml before stereotaxic delivery into the mouse brain. The animals were allowed to recover for three weeks before telemetric transmitter implantation and measurement. The circadian temperature cycle was used to assess whether animals had recovered from surgery and viral injections. All stereotactic injection sites were verified by immunohistochemistry.

To evoke hypothermia, the Vglut2-Cre mice were injected i.p. with 0.3 mg/kg CNO (BML-NS105-0025, Enzo, diluted in saline) in the morning. The $T_{core}$ was monitored closely to determine $T_{core}$ reduction soon after CNO injection. Controls were injected with equivalent volumes of saline solution. The mice were then subjected to physiological and metabolic assessments to assess the establishment of a torpor-like state.

### DBS electrodes construction

To answer the question whether WSNs in the MPNs can be activated by DBS to induce hypothermia, bipolar DBS electrodes were constructed with Teflon-coated nichrome wires (12.5 μm diameter). The two anode and cathode wires were tightly twisted together with each other, and the two exposed tips were horizontally separated by 0.05–0.10 mm. Two pairs of DBS electrodes were attached to a female connector by soldering them together and fixed with a 2 mm thick resin support base to form a bipolar DBS electrode. The remaining wires (about 8 mm) were straightened via an electrode instrument and spaced at 0.6 mm, 1.6 mm or 2.4 mm apart. The electrode wires were covered with polyethylene glycol (PEG) to maintain the linear pattern and rigidity.

### DBS surgery and electrode implementation

Mice (6–8 weeks of age) were fasted overnight but freed to drink before the surgery. Mice were injected with meloxicam (5 mg/kg, s.c.) 30 min prior to surgery and fixed on the stereotaxic apparatus under anaesthesia with 1.5% isoflurane in N2:O2 mixture during surgery using an isoflurane vaporiser (Product #R540, RWD Life Science, Shenzhen, China). Body temperature was maintained at $36.5 \pm 0.5\,°C$ with a heating blanket (Product #TCAT-2DF, Harvard Apparatus, USA). The mouse skull was exposed, and a hole was drilled using a dental drill (Product #78001, RWD Life Science) guided by stereotaxis (Product # 68861 N, RWD Life Science) to position the electrodes to target the bilateral MPNs (coordinates from Bregma, ML: ± 0.3 mm, AP: + 0.15 mm, and DV: −5.15 mm). The stereotaxic coordinates for the bilateral LPO (ML: ± 0.8 mm, AP: + 0.15 mm and DV: −5.15 mm), the VP (ML: ± 1.2 mm, AP: + 0.15 mm and DV: −5.15 mm), and the VDB (ML: ± 0.3 mm, AP: + 0.74 mm and DV: 5.15 mm) were also used for DBS as determined using the mouse brain atlas (http://labs.gaidi.ca/mouse-brain-atlas/?ml=&ap=&dv=).

Because the DBS electrode wires were thin and soft, the covering PEG was dissolved slowly using saline to expose a small piece of the wires, guided using a stereomicroscope until the wires were inserted into the correct depth to reach the medial POA bilaterally. The remaining PEG was dissolved completely, and the electrode was secured using dental cement and two stainless-steel screws threaded into the skull. More dental cement was also used to cover the space between the skull and the bottom half of the female connectors. Once the dental cement hardened, the skin edges were attached to the dental cement surface using tissue adhesive (1469SB, 3 M™ Vetbond™ Tissue Adhesive, 3 M). The mice were kept in the intensive care cage for postoperative care.

### DBS experimentation

After recovery for 3 d, the implanted stimulating electrodes were connected with a programmable eight-channel pulse stimulator (Master-8, MicroProbes, USA). The HFS output was set at continuous stimulation at 100 Hz, 1–8 V, 90 μs pulse duration. The core body temperature of mice was recorded simultaneously via rectal thermometry.

For experimentation of DBS with $T_{core}$ maintained at normothermia, the tMCAO mice were subjected to DBS (4 V and 100 Hz) after reperfusion for 1 h. The mouse's rectal temperature was continuously monitored. Once the core temperature declined from $36.5 \pm 0.5\,°C$ to $35 \pm 0.5\,°C$, a heating lamp was used to warm the mice to maintain the $T_{core}$ at $36.5 \pm 0.5\,°C$. After 4 h DBS, the $T_{core}$ was checked and kept stable at $36.5 \pm 0.5\,°C$ until the time mice were killed for TTC staining to confirm ischaemic brain damage.

### Confirmation of DBS electrode site

To confirm the anatomical location of DBS electrode, the mouse was anaesthetised with a mixture of 3% isoflurane in N2:O2 (70:30; flow rate 400 ml/min) after DBS. The electrode location was determined using an electrical lesion by passing a 20 μA current for 15 s with a lesion-making device (53500, Ugo Basile company, Italy). The mice were then transcardially perfused with 0.01 M PBS, followed by 4% freshly made paraformaldehyde under deep anaesthesia. After brain fixation, dehydration, and section, the brain section (16 μm thickness) containing the lesion areas was stained with crystal violet staining (0.5% crystal violet in 20% alcohol solution containing 660 μl acetic acid) for 8 min, and then cleaned with 100% alcohol twice and immersed in xylene. The slides were sealed with neutral resin. Brain images were captured using a microscope (Tissue FAXS Plus, Meyer Instruments, Inc., USA).

### Measurements of oxygen consumption using indirect calorimetry

Information on oxygen consumption ($VO_2$), carbon dioxide production ($VCO_2$), heat, and respiratory exchange ratio (RER) were measured and collected using the Columbus Instruments Comprehensive Lab Animal Monitoring System (CLAMS) at the Peking University Shenzhen Graduate School Animal Centre Metabolic Core. Mice were placed into metabolic cages using a computer-controlled open-circuit system. The individual mouse was acclimated, singly housed, and standard diet–fed, enabling the measurement of metabolic parameters. After adaptation in the chambers, the metabolic baseline of mice was determined. Then, hypothermia was achieved either through the injection of CNO into the Vglut2-Cre mice or by spraying water on the surface of the mice. DBS electrodes in mouse brains were connected with the stimulator. The core body temperature of each mouse was monitored via rectal thermometry. The threads of thermocouples or electrodes attached to the mouse were extended to the outside of the metabolic chamber through a small conduit sealed with adhesive tapes. After 20 min of hypothermia induction, recording started.

### Determination of blood glucose and noradrenaline levels

After 1 h reperfusion following tMCAO, mice were subjected to hypothermia for 4 h achieved by surface cooling, chemogenetic induction, or DBS stimulation. Then the blood was obtained by pricking the tail. The blood glucose level was measured using a glucometer (Roche, Accu-Check, USA). Additionally, the blood (100–200 μL) was obtained by penetrating the retro-orbital sinus in mice with a sterile haematocrit capillary tube and placed in an anticoagulant tube. After centrifugation at 3000 rpm for 5 min, the plasma was used for the detection of the noradrenaline level with an ELISA kit (CSB-E07870m, CUSABIO, China), following precisely the manufacturer's recommendations.

### Immunofluorescence staining

Mouse brains were collected after 1 h DBS followed by 1 h rest. Mice were transcardially perfused with 0.01 M PBS, followed by 4% freshly made paraformaldehyde under deep anaesthesia. The brains were fixed overnight in 4% PFA at 4 °C. After dehydration in 10% and 30%

sucrose solution, the brains were cut into 16 μm thickness sections with a cryostat microtome (CM1950, Leica, Germany).

The brain sections underwent antigen retrieval[64], followed by incubation with 5% BSA for 1 h to block the unspecific binding of antibodies. The brain sections were incubated with the primary antibodies, including mouse anti-c-Fos (ab208942, Abcam, USA, used at 1:500) and rabbit anti-BDNF (ab108319, Abcam, USA, used at 1:300) in 3% BSA in a humidified chamber overnight at 4 °C. After washing with PBS, the brain slices were incubated with Alexa Fluor 594-conjugated goat anti-mouse secondary antibody (ab150116, Abcam, USA, used at 1:500) or with an Alexa Fluor 488-conjugated goat anti-rabbit IgG (ab150077, Abcam, USA, used at 1:500). Slides were covered with a coverslip under the Fluoroshield, an aqueous anti-fade mounting medium spiked with DAPI (ab104139, Abcam, USA). Images were acquired using a fluorescence microscope (M2, Zeiss, Germany; ZEN V2.6) or a confocal microscope (LSM 710; Zeiss, Germany) and quantified using Image J.

### Fluoro-Jade B staining

Fluoro-Jade B (FJB) staining was performed to visualise degenerating neurones as previously described[65]. Brain tissue sections were cut on a cryostat microtome (CM1950, Leica, Germany) at a thickness of 16 μm. Sections were collected in 0.1 M PBS and mounted on 2% gelatin-treated microscope glass slides. All tissue slices were dried at 50 °C for at least 0.5 h. Tissue slices were first immersed into a solution of 100% EtOH for 5 min, followed by rinsing in 70% ethanol and distilled water for 2 min each session. After incubating in 0.06% potassium permanganate solution for 15 min, slides were washed with distilled water and transferred to a 0.001% solution of Fluoro-Jade B (AG310, Sigma-Aldrich, USA) dissolved in 0.1% acetic acid, Millipore, USA) for 30 min. Finally, the stained slides were washed in distilled water and dehydrated thoroughly in a heated incubator at 50 °C for 5 min. Slides were then cleared in xylene for 2 min and mounted using a coverslip under DPX mounting medium (Thermo Scientific, USA). FJB-positive cells were reported as the number of FJB-positive cells per field region.

### RNAscope fluorescent multiplex in situ hybridisation

After DBS stimulation for 1 h and rest for 1 h, the mouse brain was collected and immediately frozen in liquid nitrogen. Fluorescent in situ hybridisation was used to detect gene expression of *c-Fos* and *Adcyap1* on 20 μm cryosections of fresh frozen brain tissue after DBS. Transcripts were detected with the hybridisation probes from Advanced Cell Diagnostics Inc (CA, USA). The staining procedure was followed exactly as per the manufacturer's instructions for the RNAscope Fluorescent Multiplex Reagent V2 Kit (Cat. No. 323100).

### Acute POA brain slices preparations

Mice (either C57BL/6J or Vglut2-Cre injected with Gq-DREADD-AAV) were anaesthetized with 100 mg/kg pentobarbital sodium and decapitated. The method to produce acute brain slices was as previously described[65]. Brains were immediately dissected out and immersed in an ice-cold cutting solution containing (in mM): 30 NaCl, 26 NaHCO3, 10 D-glucose, 4.5 KCl, 1.2 NaH2PO4, 1 MgCl2, 194 sucrose, adding 1.5 mL 1 M HCl per 1 L cutting solution and bubbled with 95% O2/5% CO2. Coronal slices containing the POA were cut using a vibratome (VT1120S, Leica Systems). The slices were at 300 μm thickness for electrophysiology recording and 400 μm thickness for two-photon imaging. Brain slices were rinsed twice in artificial cerebrospinal fluid (aCSF) containing (in mM): 124 NaCl, 26 NaHCO3, 10 D-glucose, 4.5 KCl, 1.2 NaH2PO4, 1 MgCl2, 2 CaCl2, adding 10 g sucrose and 1 mL 1 M HCl per 1 L aCSF and bubbled with 95% O2/5% CO2. Slices were gently moved to a brain slice keeper containing aCSF saturated with 95% O2/5% CO2 (v/v). Incubation of slices was at 34 °C for 30 min before transferring to room temperature for at least 1 h prior to recording or imaging. Slices were then individually transferred to a recording chamber (RC26G, Warner Instruments, USA) fixed to the x–y stage of an upright microscope (BX51W, Olympus, USA). During recording or imaging, slices were continuously perfused with aCSF saturated with 95% O2/5% CO2 (v/v) at a rate of 3 ml/min. The holding chamber and tubing were fitted with a controlled heating unit to raise the temperature to 33–39 °C to activate the warm-sensitive neurones.

### Electrophysiological recordings in acute brain slices

We performed whole-cell patch-clamp recordings from acute brain slices. The spontaneous firing was recorded in the current clamp to examine neuronal excitability. The membrane potential was hold at −40 mV. Recording pipette was filled with (in mM): 128 potassium gluconate, 10 NaCl, 10 HEPES, 0.5 EGTA, 2 MgCl2, 4 Na2ATP, and 0.4 NaGTP. The acquisition frequency was at 20.0 kHz. After a 2 min recording of spontaneous firing, we used a metal electrode (Cat# 300011, FHC, Bowdoin, USA) to stimulate the POA under 0.5 V, 1 V, 2 V, 4 V, and 8 V at 100 Hz in sequence. The stimulating electrode was placed at a distance between 100–200 μm to the recording electrode. Each HFS lasted for 1–2 min, and the action potential of neurones was recorded simultaneously.

In another experiment, the excitatory and inhibitory transmissions were blocked using a cocktail of blockers including 50 μM D-AP5, 10 μM CNQX, and 20 μM bicuculline. The N-type voltage-gated calcium channel (N-VGCC) blocker ω-conotoxin GVIA (2.5 μM) was also used. The AP firing rate under the stimulation of 4 V at 100 Hz was then recorded. To confirm that the neurones recorded were indeed WSNs, we first recorded the AP firing rate at 33 °C, then stimulated the same neurone at 33 °C with 4 V 100 Hz. The same neurone was further treated with the raised temperature to 39 °C, and the AP firing was recorded without HFS. According to previous reports, a neurone was classified as warm-sensitive if it had a positive thermal coefficient that was at least +0.8 impulses $s^{-1}$ $°C^{-1}$, or moderately warm-sensitive neurones (MSNs) with a TC in between +0.2 and +0.8, and ISN with TC below +0.2[44,45]. The silent neurons were temperature insensitive. This way, neurones were confirmed to respond to both warm sensing and DBS stimulation.

Neurone spiking traces under HFS were imported into the Fitmaster software (HEKA Elektronik, Germany), and the number of action potentials was measured automatically. Neurone spiking traces in resting state or under 39 °C were exported as Igor Pro files and converted into ABF files, which were imported into Minianalysis (Synaptosoft, USA). The number of action potentials was measured automatically. Representative traces of AP firing patterns were selected for presentation, and firing frequencies were calculated.

### Calcium imaging of the warm-sensitive neurones

Acute coronal brain sections through the medial POA were cut at 300 μm in thickness and subjected to calcium imaging. Fluo-8-AM (Cat# 21080, AAT Bioquest, USA) was loaded into cells at room temperature for 20 min[58,66,67]. Fluo-8 fluorescence was measured with Ex/Em = ~490/~520 nm selected by a DG-5 system (Sutter Instrument Company, Novato, CA) and imaged with a fluorescence microscope (Leica DMI4000B, Germany). The brain slice was placed in a submerged recording chamber and perfused with oxygenated aCSF at a flow rate of 4–5 ml/min at room temperature (-25 °C). To prevent the tissue from moving in the fluid stream, the slices it was held under a metal grid (made of gold thread wire). Image time series were acquired with water-immersion objectives at different rates ranging from 0.3–3 Hz. Regions of interest within 100–150 μm of the stimulating electrode were selected. Fluorescence changes were captured using an onboard camera. Acquisition of 5 min time-lapse sequences were made, and alterations in fluorescence as a function of time were measured at a single wavelength for Fluo-8[68]. All analysis and processing were made using ImageJ/FIJI software (https://imagej.nih.gov/ij/). To visualise the spatial and temporal changes in calcium, regions of

interest over the field of view were selected, and the mean pixel intensity at each frame was measured. The data was first plotted as fluorescence intensity versus time (z) and subsequently converted to a normalised scale (ΔF/F baseline), which allows comparison across slices with the same threshold value.

The relative changes in fluorescence were calculated and normalised to the baseline measurements as $dF/F0 = (F(t) − F0)/F0$. $F(t)$ is the fluorescent value at a given time, and $F0$ is the average resting fluorescence value of baseline taken from 20 s before stimulation. The threshold was determined as $F0$ plus three times the SD of the normalised baseline intensities before stimulation. The amplitude of the peak $Ca^{2+}$ transient was the normalised amplitude value of the $Ca^{2+}$ transients greater than the threshold within the selected measurement time window. The percentage of cells responsive to 39 °C and electrical stimulation were measured using Image J automated counting.

### Visualisation of synaptic vesical release: loading

To visualise synaptic vesicle release, presynaptic boutons were loaded with 5 μM FM1-43 (Molecular Probes, Eugene, OR) in 45 mM $K^+$ aCSF for 15 min with D-APV (50 μM) added to the external solution to prevent synaptic-driven action potentials from accelerating dye release[65,69,70]. After dye-loading, slices were transferred to aCSF containing ADVASEP-7 (0.1–0.15 mM) for 30–35 min to remove any dye bound to extracellular tissue. The stimulus-induced distaining was evoked using 1–4 V at 100 Hz, 90 μs interval stimulation. The tungsten electrode was placed into the MPN region at a depth of approximately 50–100 μm. During this process, the D-APV was always in the external solution to prevent excitotoxic damage and the induction of plasticity caused by synaptic stimulation.

### Visualisation of synaptic vesical release: imaging

A two-photon patch-clamp system (Scientifica, Hyperscope, MMTP-3000, UK; HEKA, EPC-10 USB Double, GER) equipped with a Ti:sapphire laser (900 nm; Coherent, Chameleon Ultra II) was used for two-photon excitation microscopy as previously described[65]. Images were acquired using a 16 ×0.9 NA water-immersion ultraviolet objective (CFI75 LWD) with Lasersharp software provided by BioRad. A series of 6 images (512 × 512 pixels, 0.077 μm/pixel in the x–y axes) at different focal planes were acquired every 26–27 s with a 1-μm step in the z-direction. Then a subset of 3–6 contiguous in the z-axis series was selected and aligned with the first z-series by shifting each image in 3 dimensions based on the location of the peak of their cross correlograms with the first z-series. The puncta (4–5 in each slice) located 50–60 μm of the stimulating electrode and stimulus-dependent unloading were selected according to the following three criteria: first, the fluorescence intensity of more than two SD above the mean background; second, a diameter between 0.3 to 1.8 μm; and third, a roughly circular shape. Fluorescence measurements were analysed using Image J by spatially averaging signals over a region centred over each punctum for each time point during the unloading protocol. The activity-dependent destining time courses were generated by each punctum subtracting its residual fluorescence intensity (<10% of initial intensity), then normalised to the maximal fluorescence intensity of the punctum that was in the unloading procedure. The inverse halftime of decay of intensity during unloading ($1/t_{1/2}$) was then calculated.

### Statistical analysis

All analyses were performed using GraphPad Prism (Version 9.0.0, San Diego, CA). All data were presented as mean ± standard error of the mean (s.e.m.). The tMCAO surgery and hypothermia induction methods were used as two main factors for the 2ANOVA analysis. Multiple group comparisons using a 1ANOVA and 2ANOVA were followed by a *post hoc* Tukey's, Sidak's, or Dunnett's test to identify significant groups as indicated in the figure legends. An ANOVA with repeated

measures (RM-1ANOVA) was performed on neurones subjected to a series of voltages of HFS and subsequent thermal stimulations.

Statistical details for specific experiments - including exact n values and what n represents, precision measures, statistical tests used, and definitions of significance - can be found in figure legends. Where representative images were shown, each experiment was repeated at least three times independently with similar results. A $P$ value < 0.05 was taken to indicate statistical significance. 1ANOVA: one-way ANOVA; 2ANOVA: two-way ANOVA; RM-1ANOVA: an ANOVA with repeated measures; #$P$ < 0.05; ##$P$ < 0.01; ###$P$ < 0.001, ####$P$ < 0.0001.

### Reporting summary

Further information on research design is available in the Nature Portfolio Reporting Summary linked to this article.

## Data availability

The data generated in this study are provided in the Supplementary Information/Source Data File. Source data are provided with this paper.

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

## Acknowledgements

We thank the SUSTech Animal Facility for animal maintenance, the SUSTech Core Facility for technical support, and the Peking University Laboratory Animal Centre of Shenzhen Graduate School for the instruction on the use of the metabolic cages. Financial supports for S.-T.H. were from grants from the National Natural Science Foundation of China (81871026); the Shenzhen-Hong Kong Institute of Brain Science-Shenzhen Fundamental Research Institutions (2021SHIBS0002, 2022SHIBS0002); Shenzhen Science and Technology Innovation Committee Research Grant (KQJSCX20180322151111754, JCYJ20180504165806229). S.-T.H. is also supported by the Guangdong Innovation Platform of Translational Research for Cerebrovascular Diseases and SUSTech-UQ Joint Centre for Neuroscience and Neural Engineering (CNNE).

## Author contributions

S.Z. and X.P.Z. performed mouse tMCAO surgery, behavioural tests, stereotactic injections, DBS, immunofluorescence staining, histopathological evaluation, mouse brain and recordings for body temperature, metabolic rates, and data analysis; B.L. helped with tMCAO surgery and DBS; Y.J.W. for Vglut2-Cre mice feeding, virus injection and in situ RNAscope staining; X.Y.L. and J.J. performed brain slice electrophysiology and data analysis; H.L.Z. performed presynaptic vesicle release experiments, calcium imaging in brain slices, and Z.Y.Z. analysed the calcium data; H.Y. performed BDNF immunostaining; K.S. supervised the hypothermia-induction experiment using Vglut2-Cre mice and participated in the initial planning of the project; Y.Z.W. provided critical discussions and revised the draft; S.-T.H. conceived the idea, secured funding, designed the experiments, analysed the results, revised the first draft, and wrote the manuscript.

## Competing interests

The authors declare no competing interests.
