## [Peer Review File · Nature Communications]

Hypothermia evoked by stimulation of medial preoptic nucleus protects the brain in a mouse model of ischaemiaReviewer #1 (Remarks to the Author):

This paper investigates the idea that deep brain stimulation (DBS) in the medial preoptic area (MPO) can be used to lower body temperature and thereby prevent brain tissue damage associated with ischaemia. A strength of the manuscript is a careful side-by-side comparison of how surface cooling and chemogenetic stimulation of warm-sensitive neurons (WSNs) differentially effect a wide range of outputs (Fig. 2). The differences are striking and this data will be useful to the field. A second strength is the demonstration that DBS in a mouse can phenocopy WSN stimulation or MPO warming. This suggests that DBS could be a useful strategy to lower body temperature therapeutically.

I have a few recommendations for revision.

Major points

- 1. The paper starts with an extended investigation of how the tMCAO model affects body temperature rhythms, but it is never explained why this analysis is being performed (or what the cause of any observed changes in body temperature might be). If the reason for this experiment is simply to demonstrate that tMCAO does not itself cause therapeutic levels of hypothermia, then this control could be briefly described later in the paper. As it stands, it is confusing to start with an extensive discussion of supplementary data that is ancillary to the main point of the paper.**
- 2. The Fos experiments in Extended Figure 8 are central to the paper, since this is the only direct evidence that WSNs are activated by DBS in vivo. The authors should quantify the colocalization between Fos and *Adcyap1* and *Bdnf* (and vice versa) following DBS. They should also provide more comprehensive documentation and quantification of where exactly in the MPO Fos is induced by DBS. From the pictures in Extended Figure 8, most of the Fos appears to be located lateral to the traditional VMPO/MnPO region where warm-responsive neurons have been identified. If this is the case, that is fine, but it needs to be documented.**
- 3. How were the neurons selected for recording in Fig. 4e,f? It is surprising if 57% of random cells (or even *Vglut2+* neurons) are intrinsically warm sensitive. Second, more than two data points are needed to define a line of best fit for the temperature coefficient (Fig. 4f). Third, it is not appropriate to exclude cells because they are silent at 26 C, when the experiment is measuring temperature responses up to 39 C. The responses of the "silent cells" to heating should be shown.**
- 4. The MPO is a large structure, and there is no data presented showing how placement of the DBS electrode affected the body temperature responses. A figure showing the locations of the electrode placements and the relative response should be included in the supplement. Ideally, there would also be some data showing that stimulation of an adjacent structure does not influence body temperature.**
- 5. There are previous studies showing that electrical stimulation of the POA induces heat loss responses. They should be cited and discussed.**

e.g.

Andersson BR et al., Central control of heat loss mechanisms in the goat, 1956

Hemingway A., et al., Shivering suppression by hypothalamic stimulation, 1954

Minor points

- 1. Mouse gene names are in italics with the first letter capitalized.**
- 2. There is no such thing as c-FOS. The mouse gene is *Fos* (in italics).**
- 3. A compelling experiment would be to silence WSNs during DBS (e.g. with inhibitory DREADDs) and show that this blocks the body temperature decrease.**
- 4. It is mentioned that DBS treated mice compared to untreated mice have reduced VO_2 , RER and Heat, but the traces for the control animals are not shown (Fig 3i-k). This should be included.**
- 5. The purpose of the purple squares in Fig. 3 H-K is confusing -- it should be labelled in the figure that it represents the time of stable BT at 33 C.**
- 6. Figure 4g is confusing. This needs to be explained better in the legend and text.**

Reviewer #2 (Remarks to the Author):

Excellent manuscript with detailed description of the chemical and surface cooling with thoroughly explored outcomes and temperature monitoring. I would like to see a better description of the certainty that the DBS truly was in the medial preoptic nucleus or discuss possible neighboring effects.

Reviewer #3 (Remarks to the Author):

The authors submit an interesting and thought-provoking manuscript on the possible development of a DBS-based intervention to produce hypothermia and reduction of metabolism for neuroprotection. The concept is very interesting and pioneering and the authors have conducted extensive experimentation for proof-of-principle as well as mechanistic examination.

The following are points of concern or for clarification:

1. **Language.** The language is sometimes unclear throughout the manuscript, in particular the abstract. A review by an English expert will improve the experience of the readers if the manuscript is published.
2. **Paragraph starting in line 96.** The Tcore change is reported to change by 2 degrees plus or minus. This is confusing. Was this meant to be a comparison with Tcortex?
3. **The motor outcome measures chosen by the authors tend to be poorly reproducible compared to skilled reaching tasks. Why were these metrics chosen?**
4. **The authors show with in-vitro (slice) physiology that MSNs can be modulated pharmacologically and electrically. The effects, overall, seem similar. Wouldn't it be better to evaluate if the effects of DBS on metabolism could be blocked, in vivo, by drugs known to block MSNs? Wouldn't that be a more direct mechanistic examination on the effects of mPON DBS? Is that not possible or are those agents too toxic for in-vivo use?**
5. **DBS of mPON seems to have produced an effect on metabolism, temperature and behavioral function. The animals were in a state of sedation. If this is the case, what would be the advantage of PON DBS over drugs that are known to reduce metabolism such as barbiturates?**
6. **In the same line, why focus on the effects of DBS on body temperature? Isn't the effect really on metabolism and the temperature is just a consequence of reduced metabolism?**
7. **The authors tested the effects of mPON DBS as well as the chemogenetic intervention on stroke volume and motor function. Were these animals randomized to treatment group? If not, how do the authors account for the random variability in stroke size and motor deficits that is inherent to these stroke models?**
8. **The authors measured the cortical temperature and core temperature in a group of animals but not in the DBS-treated animals. Why is that? The animals already had brain instrumentation and it would not be difficult to add the cortical thermocouple probe.**

Reviewer #4 (Remarks to the Author):

The study described in this manuscript investigated how hypothermia evoked by hypothalamic warm-sensitive neurons protects the ischemic brain. The stated aim of this study was to establish and develop a new hypothermic method in acute stroke. The authors conclude that a successful clinical translation of deep brain stimulation (DBS)-evoked hypothermia would benefit stroke patients. This study deals with an interesting, potentially important topic that fits well in the scope of this journal. However, there are some shortcomings that need to be addressed before any consideration in publication in this prestigious journal.

In the abstract, more information needs to be provided, including the objectives,

methods, results and conclusion. Particularly, what species was utilized in the experiments? What are the observed percentage changes of the measurement for the different treatments? It would be of interest to have key data presented in the abstract given as mean +/- SEM, n=, p<, etc. to allow the readers to evaluate the data upfront. The remarks in the lead sentence of the abstract about the inadequacy of aspirin or heparin for reducing blood clots or hemolytic products for dissolving blood clots are not relevant to this study.

In the introduction, the authors need to thoroughly address the rationale and significance of the present study, especially regarding induced hibernation in mammals, and how they developed the hypothesis that DBS-evoked hypothermia through WSNs could overcome the side effects of cold sensing-triggered hypothermia and could be used to protect against ischemic brain injury. The new technology should be validated.

In the materials and methods, the authors need to clearly address and clarify the use of the chemogenetic approach to specifically activate excitatory neurons (including WSNs) in the medial preoptic nucleus in order to induce hypothermia, and the use of DBS electrodes. According to the authors, the DBS-evoked hypothermia is new, but they focus more on the effects of the chemogenic-evoked and DBS-evoked hypothermia techniques rather than elucidating the mechanism of action. We need more mechanistic information to comprehensively understand the reality of the technique.

In the results, the authors assume that the reduced metabolic rates was due to chemogenetic-evoked hypothermia, but the cause-and-effect relationship was not confirmed. The authors discuss several possible pieces of evidence showing that WSN is the key for DBS-triggered hypothermia; however, direct evidence is needed.

The manuscript could benefit from another round of editing to correct minor mistakes and improve clarity. For example, "chemogenetic" is spelled inconsistently.

Point-by-point Reply (answers in blue colour)

Reviewer #1 (Remarks to the Author):

This paper investigates the idea that deep brain stimulation (DBS) in the medial preoptic area (MPO) can be used to lower body temperature and thereby prevent brain tissue damage associated with ischaemia. A strength of the manuscript is a careful side-by-side comparison of how surface cooling and chemogenetic stimulation of warm-sensitive neurons (WSNs) differentially effect a wide range of outputs (Fig. 2). The differences are striking and this data will be useful to the field. A second strength is the demonstration that DBS in a mouse can phenocopy WSN stimulation or MPO warming. This suggests that DBS could be a useful strategy to lower body temperature therapeutically.

Answer: Thank you for spending the time reviewing our manuscript. We appreciate your positive comments and constructive suggestions to improve the paper. We have now added new experiments to address all your concerns. The manuscript was revised accordingly and is now significantly improved. Thank you!

I have a few recommendations for revision.

Major points

1. The paper starts with an extended investigation of how the tMCAO model affects body temperature rhythms, but it is never explained why this analysis is being performed (or what the cause of any observed changes in body temperature might be). If the reason for this experiment is simply to demonstrate that tMCAO does not itself cause therapeutic levels of hypothermia, then this control could be briefly described later in the paper. As it stands, it is confusing to start with an extensive discussion of supplementary data that is ancillary to the main point of the paper.

Answer: Thank you for raising this important point. The reviewer is correct that the reason for this experiment is to demonstrate that tMCAO did not itself cause therapeutic levels of hypothermia. We agree with the reviewer's suggestion and have moved this section to Line 165, a location just before the section "DBS of the MPN evokes deep hypothermia" as an ancillary to the main point of the paper.

2. The Fos experiments in Extended Figure 8 are central to the paper, since this is the only direct evidence that WSNs are activated by DBS in vivo. The authors should quantify the colocalization between Fos and Adcyap1 and Bdnf (and vice versa) following DBS. They should also provide more comprehensive documentation and quantification of where exactly in the MPO Fos is induced by DBS. From the pictures in Extended Figure 8, most of the Fos appears to be located lateral to the traditional VMPO/MnPO region where warm-responsive neurons have been identified. If this is the case, that is fine, but it needs to be documented.

Answer: Thank you for raising these important questions. We agree with the reviewer's comments. To this end, we have now performed new experiments (Fig. 5) to show quantifications of the co-localization of c-Fos with BDNF (protein), *Adcyap1* (mRNA) (and vice versa) using double immunohistochemistry and RNAscope *in situ* labelling, respectively. DBS activation of c-Fos and *Adcyap1* were shown on coronal sections cut at two Bregma points (+0.15 and 0.26 mm) to indicate the c-Fos expressions in relation to the DBS electrode (Fig. 5a, c). Based on these new data, c-Fos was mostly expressed in the medial preoptic area (MPA), including the MPN, and less in the lateral preoptic area (LPO). Based on these new data, we determined that the DBS site was in the MPN.

3. How were the neurons selected for recording in Fig. 4e,f? It is surprising if 57% of random cells (or even Vglut2+ neurons) are intrinsically warm sensitive.

Answer: Thank you, and sorry for the confusion. In the original manuscript Fig. 4e, f, neurons in the medial preoptic nucleus region on brain slices were first sampled electrophysiologically to detect spontaneous firing. If the neuron's AP firing rate was zero, it was regarded as a silent cell. Then the clamped neurone was further subjected to increasing temperatures from 26, 33, to 39 °C to record the increased firing rate as a measure to determine warm sensitivity. The warm-sensitive neurons (WSN, TC > 0.8) and the moderately warm-sensitive neuron (MSN, TC between 0.2 and 0.8) were plotted together and shown in Fig. 4f. While the temperature-insensitive neuron (ISN, TC < 0.2), and silent neurons were NOT plotted in the original Fig 4f. Since the silent cells don't respond to increasing temperature, we did not consider silent cells.

To clarify this, we re-performed the experiment to add new data from ISN and silent cells, as shown in the revised Fig. 6f (shown below). This allowed us to recalculate and

present all portions of neurons responsive to the increasing temperatures. As shown in the pie chart Fig. 6g (below), WSN = 20%, MSN = 22.5%, ISN = 40%, silent cells = 17.5%.

Our findings are in agreement with a recent report by Kamm GB et al. (2021) PMID: 34672983 (Figure E below). They reported the proportion of WSN, MSN, ISN and silent cells in the POA as 27:23:33:16%, respectively.

(E) Distribution of POA cell categories based on their electrophysiologically recorded temperature sensitivities obtained from Trpm2^{+/+} (left) and Trpm2^{-/-} (right) slices. Warm-sensitive neuron (WSN), moderately warm-sensitive neuron (MSN), temperature-insensitive neuron (ISN), cold-sensitive neuron (TC < -0.6; CSN), and silent neuron firing no APs (silent) are shown. (From

Kamm GB et al. PMID: 34672983)

Second, more than two data points are needed to define a line of best fit for the temperature coefficient (Fig. 4f).

Answer: Thank you. We have re-performed the experiment to add a new temperature point at 36 °C in the revised Fig. 6f. The new data supports our original conclusion.

Third, it is not appropriate to exclude cells because they are silent at 26 C, when the experiment is measuring temperature responses up to 39 C. The responses of the "silent cells" to heating should be shown.

Answer: Thank you for raising this important question. We have re-performed the experiments depicted in the original Fig. 4f. The new data were added to the revised Fig. 6f. The silent cells don't respond to increasing temperature. Therefore, the conclusion drawn from the original data shown in Fig. 4f remains unchanged.

4. The MPO is a large structure, and there is no data presented showing how the placement of the DBS electrode affected the body temperature responses. A figure showing the locations of the electrode placements and the relative response should be included in the supplement. Ideally, there would also be some data showing that stimulation of an adjacent structure does not influence body temperature.

Answer: Thank you. First, we have added data from experiments using the electrical lesion method to mark the locations of the stimulation electrodes (red colour arrows) for MPN (Fig. 3b'), LPO, VP (Supplementary Fig. 6c), and VBD (Supplementary Fig. 6d). These figure panels are shown below.

Figure 3a, b, b' (below)

Supplementary Fig. 6c,d (below)

Second, these sites were stimulated with DBS to determine the effect of Tcore. As shown in the new Fig. 4g, j (shown below), DBS electrodes were placed bilaterally into MPN (Bregma ML/AP/DV: $\pm 0.3/+0.15/-5.15$ mm), LPO (ML/AP/DV: $\pm 0.8/+0.15/-5.15$ mm), and VP (ML/AP/DV: $\pm 1.2/+0.15/-5.15$ mm). Another DBS site was VDB (ML/AP/DV: $\pm 0.3/+0.74/-5.15$ mm). Except for the MPN site, stimulation of these sites did not produce deep hypothermia. Although there occurred shallow reductions of Tcore during VDB and LPO stimulations, the Tcore did not drop below 35 °C. Based on these new data, we determined that the ideal DBS site is in the MPN. These new data have been added to the new Fig. 4.

5. There are previous studies showing that electrical stimulation of the POA induces heat loss responses. They should be cited and discussed.

e.g.

Andersson BR et al., Central control of heat loss mechanisms in the goat, 1956

Hemingway A., et al., Shivering suppression by hypothalamic stimulation, 1954

Answer: Thank you. These references have been cited and discussed in the revised manuscript, Line 392-400.

Minor points

1. Mouse gene names are in italics with the first letter capitalized.

Answer: Thank you. We have corrected all the errors following the "Guidelines for Nomenclature of Genes, Genetic Markers, Alleles, and Mutations in Mouse and Rat" (<http://www.informatics.jax.org/mgihome/nomen/gene.shtml>). We revised the gene and protein symbols: gene symbols are italicized, with only the first letter in upper-case (e.g., *c-Fos*, *Adcyap1*). Protein symbols are not italicized, and all letters are in upper-case (e.g., BDNF).

2. There is no such thing as c-FOS. The mouse gene is *Fos* (in italics).

Answer: Thank you. We have corrected all the mistakes mentioned above.

3. A compelling experiment would be to silence WSNs during DBS (e.g. with inhibitory DREADDs) and show that this blocks the body temperature decrease.

Answer: Thank you. We have performed the required experiments using AAV2/9-hSyn-DIO-hM4D(Gi)-eGFP-WPRE-pA inhibitory virus, which was injected into the MPN of *Vglut2-ires-cre* knock-in and *Adcyap1-2A-Cre* knock-in mice. Silencing of excitatory *Vglut2* neurones completely blocked DBS-evoked hypothermia (Fig. 5m, below). Interestingly, inhibiting the *Adcyap1* gene in the MPN did not completely block DBS-evoked reduction in *Tcore* (Fig. 5n), suggesting the involvement of other WSNs in the MPA in response to DBS stimulation. These new data strongly support our hypothesis

that DBS activated WSNs to induce hypothermia. These data have been added to the new Fig. 5.

4. It is mentioned that DBS treated mice compared to untreated mice have reduced VO₂, RER and Heat, but the traces for the control animals are not shown (Fig 3i-k). This should be included.

Answer: Thank you. We have added the traces of control mice (untreated) VO₂, RER, and Heat in the revised Fig. 3i-k (below).

5. The purpose of the purple squares in Fig. 3 H-K is confusing -- it should be labelled in the figure that it represents the time of stable BT at 33 C.

Answer: Thank you. We have directly labelled all the purple squares in Fig. 3 with "T_{core}=33°C" and explained this in Line 1084 (Figure legends for Fig 3).

6. Figure 4g is confusing. This needs to be explained better in the legend and text.

Answer: Thank you, and sorry for the confusion. We have added detailed explanations in Line309-318.

Reviewer #2 (Remarks to the Author):

Excellent manuscript with detailed description of the chemical and surface cooling with thoroughly explored outcomes and temperature monitoring. I would like to see a better description of the certainty that the DBS truly was in the medial preoptic nucleus or discuss possible neighboring effects.

Answer: Thank you for spending time reviewing our manuscript. We appreciate your constructive comments. Your questions regarding the "certainty that the DBS truly was in the medial preoptic nucleus or discuss possible neighboring effects" are important. To address these questions, we have performed the following new experiments and added the results in the revised manuscript.

First, we did detailed quantifications of the co-localization of c-Fos with BDNF and *Adcyap1* (mRNA) (and vice versa) using double immunohistochemistry and RNAscope *in situ* labelling, respectively. The co-localizations were shown on coronal sections cut at two Bregma points (AP: +0.15 and +0.26 mm) to indicate the c-Fos expressions in relation to the DBS electrode at the medial preoptic nucleus (MPN) site (ML/AP/DV: $\pm 0.3/+0.15/-5.15$ mm) (Fig. 5a, c, as shown below). The highest intensity of c-Fos was at the medial preoptic area (MPA) and MPN (Fig. 5 h) and less in the lateral preoptic area (LPO). Based on these new data, we determined that the DBS site was in the MPN.

Second, we placed DBS electrodes bilaterally into several sites, including LPO (ML0.8 mm), VP (ML1.2 mm) with the same AP at +0.15 mm, DV at 5.15 mm of the MPN site (ML0.3 mm). Another DBS site was VDB with ML at 0.3 mm, AP at +0.74 mm, and DV at 5.15 mm (Fig. 4g-k, as shown below). Stimulation of these sites did not produce deep hypothermia as seen in MPN DBS. Although there occurred shallow reductions

of the Tcore in VDB and LPO stimulations, the Tcore did not drop below 35 °C to achieve the necessary hypothermia. These data demonstrated that MPN is a specific target nucleus for DBS hypothermia.

Third, we have added data from experiments using the electrical lesion method to mark the locations of the stimulation electrodes (red colour arrows) for MPN (Fig. 3b'), LPO, VP (Supplementary Fig. 6c), and VDB (Supplementary Fig. 6d). These figure panels are shown below.

Figure 3a, b, b' (below)

Supplementary Fig. 6c,d (below)

Together, these two experiments provided a better description of the DBS site in the medial preoptic nucleus (MPN) and established the MPN site as an effective DBS target for therapeutic hypothermia.

Reviewer #3 (Remarks to the Author):

The authors submit an interesting and thought-provoking manuscript on the possible development of a DBS-based intervention to produce hypothermia and reduction of metabolism for neuroprotection. The concept is very interesting and pioneering and the authors have conducted extensive experimentation for proof-of-principle as well as mechanistic examination.

Answer: We thank the reviewer for the positive and encouraging comments.

The following are points of concern or for clarification:

1. Language. The language is sometimes unclear throughout the manuscript, in particular the abstract. A review by an English expert will improve the experience of the readers if the manuscript is published.

Answer: We apologize for the confusion and have asked a native English-speaking colleague to proofread the manuscript, in particular the Abstract.

2. Paragraph starting in line 96. The Tcore change is reported to change by 2 degrees plus or minus. This is confusing. Was this meant to be a comparison with Tcortex?

Answer: Sorry for the confusion. We meant the fluctuation of the core body temperature (Tcore) between the Sham group and the ischaemic tMCAO group mice. This clarification has been added in the revision Line 179-183. Thank you.

3. The motor outcome measures chosen by the authors tend to be poorly reproducible compared to skilled reaching tasks. Why were these metrics chosen?

Answer: Thank you. The neurological deficit scores and forepaw pulling strength measurements are all very reproducible tests for the MCAO model of stroke. Our MCAO mouse model causes partial paralysis of the limbs on the contralateral side. The MCAO mice roll or turn to the opposite side (neurological deficit score). The pulling strength is measured using an electronic meter, producing an objective readout of the motor function deficit. These outcomes are well described in the literature (e.g. PMID: 23232947). Because of the limb motor function deficits, it is challenging and not reproducible to do skilled reaching tests and swim water maze tests. Thank you.

4. The authors show with in-vitro (slice) physiology that MSNs can be modulated pharmacologically and electrically. The effects, overall, seem similar. Wouldn't it be better to evaluate if the effects of DBS on metabolism could be blocked, in vivo, by drugs known to block MSNs? Wouldn't that be a more direct mechanistic examination on the effects of mPON DBS? Is that not possible or are those agents too toxic for in-vivo use?

Answer: Thank you for the question. The cause-and-effect of metabolism in DBS is an important one. There is no doubt that DBS of the medial preoptic nucleus causes the reduction in metabolism, but whether this is directly through the warm sensitive neurons or indirect pathways is not clear. Evaluating the direct link between DBS and metabolism is challenging in the absence of very specific drugs, which warrants further studies. Nevertheless, we designed and performed a new *in vivo* experiment to show the essential role of core body temperature in DBS brain protection:

DBS was conducted on ischaemic mice (tMCAO) while maintaining the core body temperature at normothermia (36.5°C). This way, the DBS was not effective in protecting the ischaemic brain (Fig. 4a-f, below). This data indicates that lowering the core body temperature is critical for brain protection, while the state of metabolism may be consequential. This is supported by several recent important reports of mechanisms of torpor state induction in mice which works through the lowering of the core body temperature setpoint followed by the slowdown of metabolism (PMID: 35440490; 35440490; 35440490; 35869064). In future studies, we will collaborate with expert laboratories in metabolism research to design specific ways to quickly reduce metabolism to help delineate the relationships between DBS and metabolism in brain protection. These new data have been added in Fig. 4. Thank you.

5. DBS of mPON seems to have produced an effect on metabolism, temperature and behavioral function. The animals were in a state of sedation. If this is the case, what would be the advantage of PON DBS over drugs that are known to reduce metabolism such as barbiturates?

Answer: We thank the reviewer for raising this important question. DBS mice are in a torpor-like state, not a sedated state. As shown in Supplementary Video 1, DBS hypothermic mice are fully awake, free-moving, and void of muscular deficits, indicating a torpor-like state rather than a sedated state.

DBS is advantageous compared with sedative drugs such as barbiturates. Sedative drugs often have unintended targets and side effects. Barbiturates can be extremely dangerous because the correct dose is difficult to predict. Even a slight overdose can cause coma or death. Barbiturate-mediated neuroprotection has been attributed to redistribution of cerebral blood flow to injured areas, Na-channel and glutamate receptor blockade, inhibition of calcium influx, inhibition of free radical formation, and potentiation of GABAergic activity, not to mention reduced metabolism (PMID: 6314116, PMID: 4032291; PMID: 10953037).

Barbiturates, like many other sedative-hypnotics, can also reduce core body temperature. However, within the range of safe dosage, it is not enough to produce a therapeutic level of hypothermia. For example, the lowest temperatures observed in the group of 16 patients were averaged 35.5 +/- 2.0 °C, which is not enough to achieve therapeutic hypothermia (PMID: 7273815).

Barbiturates protection against ischemic injury only if the ischemic insult is relatively mild. With moderate to severe insults, this neuronal protection is not sustained after a prolonged recovery period. Thus, barbiturates do not reduce delayed neuronal death caused by apoptosis. The long-term effects of barbiturates on cerebral ischemic injury are not yet fully defined (PMID: 15875133). Additionally, reports showed that the functional outcome of hypothermia therapy was not improved by additional barbiturate therapy (PMID: 11059666).

Nevertheless, we performed a new experiment to demonstrate the effect of barbiturates (Fig below). Pentobarbital sodium injection (20 mg/kg) reduced the Tcore (Fig below **a, b**), but the drop of Tcore was less than 35°C, and the duration was short. The level of Tcore was insufficient to produce dramatic brain protection and behavior benefits (Figure below **d-g**). Mice that received pentobarbital injection were completely void of movement and activity (Figure below **c**). These data showed that although barbiturates provided shallow hypothermia, the brain protection was not optimal compared with the DBS technique.

Figure legends:

a, Schematic of experimental design. **b**, The reduced Tcore immediately after each injection of 20 mg/kg of pentobarbital sodium. **c**, Mouse activity counts. **d**, tMCAO brain infarction measurement. **e**, edema volume measurement. **f**, neurological scores. **g**, forepaw fulling strength measurements.

In contrast, DBS-evoked hypothermia is specific to the warm sensing neurons in the POA. The degree of hypothermia is controllable throughout the process. The ideal Tcore can be maintained at 32-34 °C for a stable period, while the drug-induced hypothermia dynamics depend on the specific drug's pharmacodynamics. Therefore, using DBS to induce therapeutic hypothermia is advantageous compared with sedative drugs such as barbiturates. We have discussed this in the revision on Line 385-395.

6. In the same line, why focus on the effects of DBS on body temperature? Isn't the effect really on metabolism and the temperature is just a consequence of reduced

metabolism?

Answer: Thank you for this question. As explained in the above questions 4 and 5, lowering the central temperature control setpoint through the warm sensing neurones is a critical first step in reducing the core body temperature and the subsequent slowdown of metabolism during the torpor state. The reduced metabolic mechanisms may kick in to sustain the hypothermia and the torpor-like state. This is why we focused on the mechanisms of how DBS initiation of the lowering of Tcore.

Having said that, the hypothalamus comprises many different sub-areas involved in the regulation of temperature, metabolism, sleep, and so on, which is a complex system. The torpor induction and thermoregulatory pathways are closely intertwined, and hypothermia and hypometabolism may co-occur. It will be challenging and exciting work to distinguish body temperature and metabolism regulation in the hypothalamus in the future (PMID: 27813827: The Central Control of Energy Expenditure: Exploiting Torpor for Medical Applications; PMID: 35301430: Hypothalamic control of energy expenditure and thermogenesis; PMID: 22063719: Central control of thermogenesis). Only until then, the importance of the reduction in metabolism as a mechanism of DBS neuroprotection can be fully elucidated. Thank you.

7. The authors tested the effects of mPON DBS as well as the chemogenetic intervention on stroke volume and motor function. Were these animals randomized to treatment group? If not, how do the authors account for the random variability in stroke size and motor deficits that is inherent to these stroke models?

Answer: Thank you. The ARRIVE guideline was followed, and animal randomization was performed as described in the revised Methods section (Line 469-473). For DBS experiments, both sexes were used, including all genotypes of mice. Mice were randomly assigned into groups. For experimental stroke mice, only male mice were used to avoid the well-known fact that female experimental stroke mice tend to produce inconsistent brain infarction volume, which can be affected by the circulating blood level of estrogen and the hormonal cycles of the female mice. We have reported these in the Reporting Summary and described them in the Method section Line 469-473. The cohort of stroke mice was randomly assigned to either receive DBS and CNO injections (chemogenetic intervention) or without the treatment as controls. The inclusion criteria were that the stroke model must be successful based on the initial neurological test score of ≥ 2 at 30 min after the stroke surgery. Mice without successful stroke will be excluded from the study to minimize the possible random variability in stroke size and motor deficits. Thank you.

8. The authors measured the cortical temperature and core temperature in a group of animals but not in the DBS-treated animals. Why is that? The animals already had brain instrumentation and it would not be difficult to add the cortical thermocouple probe.

Answer: Thank you, and sorry for the confusion. We have indeed measured the cortical temperature and the core body temperature during DBS at various voltages, as shown in the original Supplementary Fig. 7 a-e. The correlation analyses showed that DBS did not alter cortical temperature before, during, and after the DBS process (Supplementary Fig. 7, below). Thank you.

Reviewer #4 (Remarks to the Author):

The study described in this manuscript investigated how hypothermia evoked by hypothalamic warm-sensitive neurons protects the ischemic brain. The stated aim of this study was to establish and develop a new hypothermic method in acute stroke. The authors conclude that a successful clinical translation of deep brain stimulation (DBS)-evoked hypothermia would benefit stroke patients. This study deals with an interesting, potentially important topic that fits well in the scope of this journal. However, there are some shortcomings that need to be addressed before any consideration in publication in this prestigious journal.

Answer: Thank you for your positive comments on the manuscript. We are encouraged by your constructive suggestions to improve this manuscript. To this end, we have restructured the manuscript, performed new experiments, and added new mechanistic data to support the conclusion. Your questions are addressed point-by-point as follows.

In the abstract, more information needs to be provided, including the objectives, methods, results and conclusion. Particularly, what species was utilized in the experiments? What are the observed percentage changes of the measurement for the different treatments? It would be of interest to have key data presented in the abstract given as mean \pm SEM, n=, p<, etc. to allow the readers to evaluate the data upfront. The remarks in the lead sentence of the abstract about the inadequacy of aspirin or heparin for reducing blood clots or hemolytic products for dissolving blood clots are not relevant to this study.

Answer: We thank the reviewer for the helpful comments. We have restructured the abstract according to the reviewer's suggestions to indicate objectives, methods, results, and conclusions clearly. We also clearly stated that these experiments were performed using mice. However, according to Nature Communications' journal instructions, as shown below, the abstract is meant to be brief and non-technical nature. We have also removed the irrelevant first sentence in the abstract.

Abstract. Provide a general introduction to the topic and a brief non-technical summary of your main results and their implication.

Text length and formatting. Attention to the following details can help expedite publication if we invite a revision after external review.

- **Articles:** an abstract of approximately 150 words, unreferenced;

In the introduction, the authors need to thoroughly address the rationale and significance of the present study, especially regarding induced hibernation in mammals, and how they developed the hypothesis that DBS-evoked hypothermia through WSNs could overcome the side effects of cold sensing-triggered hypothermia and could be used to protect against ischemic brain injury. The new technology should be validated.

Answer: Thank you. We have rewritten the Introduction to address the rationale, significance, and the way how the hypothesis was developed.

In the materials and methods, the authors need to clearly address and clarify the use of the chemogenetic approach to specifically activate excitatory neurons (including WSNs) in the medial preoptic nucleus in order to induce hypothermia, and the use of DBS electrodes.

Answer: Thank you. We have revised the Methods section (Line 590-610, Line 613-616, Line 651-663) to clarify the use of the chemogenetic approach to specifically activate excitatory neurons (including WSNs) in the medial preoptic nucleus in order to determine the torpor like state during hypothermia.

According to the authors, DBS-evoked hypothermia is new, but they focus more on the effects of the chemogenetic-evoked and DBS-evoked hypothermia techniques rather than elucidating the mechanism of action. We need more mechanistic information to comprehensively understand the reality of the technique.

Answer: Thank you for these important questions. Recently, several high-impact papers have demonstrated the critical role of POA WSNs in inducing hypothermia and torpor-like states in mice (PMID: 35440490; 35440490; 35440490; 35869064). We, therefore, focused the current study on carefully establishing the link between DBS activation of WSNs in the MPN. Our data support the hypothesis that DBS indeed can activate MPN WSNs. To provide more direct evidence to understand the reality of the DBS technique, we have performed three new experiments to determine the selectivity and specificity of DBS mechanistically. These experiments are as follows:

First, we performed detailed quantifications of the co-localization of c-Fos with BDNF and *Adcyap1* (mRNA) (and vice versa) using double immunohistochemistry and RNAscope *in situ* labelling, respectively. The co-localizations were shown on coronal sections cut at two Bregma points (AP: +0.15 and +0.26 mm) to indicate the c-Fos expressions in relation to the DBS electrode, which was at ML at 0.3 mm, AP at +0.15 mm, DV at 5.15 mm (Fig. 5a, c, as shown below). The highest intensity of c-Fos was at the medial preoptic area and medial preoptic nucleus (Fig. 5 h) and less in the lateral preoptic area (LPO). Based on these new data, we determined that the DBS site was truly in the MPN.

Second, we placed DBS electrodes at several places in the brain to determine the selectivity of DBS-induced hypothermia. DBS electrodes were placed bilaterally into LPO (ML0.8 mm), VP (ML1.2 mm) similar to the MPN site (ML0.3 mm) with AP at +0.15 mm, DV at 5.15 mm. Another DBS site used was VDB at ML0.3 mm, AP at +0.74 mm, and DV at 5.15 mm. Stimulation of these sites did not produce deep hypothermia, as seen in DBS MPN. Although there occurred shallow reductions of the core body temperature in VDB and LPO stimulations, the Tcore did not reduce below 35 °C.

Third, to unequivocally demonstrate that DBS activation of WSNs is key in inducing therapeutic level of hypothermia, inhibitory AAV2/9-hSyn-DIO-hM4D(Gi)-eGFP-WPRE-pA was injected into the MPN of *Adcyap1*-2A-Cre knock-in (B6.Cg-*Adcyap1*^{tm1.1(cre)Hze}/ZakJ) and *Vglut2*-ires-cre knock-in (C57BL/6J) mice to silence *Adcyap1* and *Vglut2* neurons, respectively. Silencing of excitatory *Vglut2* neurones completely blocked DBS-evoked hypothermia (Fig. 5m). Interestingly, inhibiting the *Adcyap1* gene in the MPN did not completely block DBS-evoked reduction in Tcore

(Fig. 5n), suggesting the involvement of other WSNs in the MPA in response to DBS-stimulation.

Together, these experiments provided a better description of the certainty that the DBS activates WSNs in the medial preoptic nucleus (MPN) and that the most effective DBS site is in MPN.

In the results, the authors assume that the reduced metabolic rates was due to chemogenetic-evoked hypothermia, but the cause-and-effect relationship was not confirmed. The authors discuss several possible pieces of evidence showing that WSN is the key to DBS-triggered hypothermia; however, direct evidence is needed.

Answer: Thank you. We agree that delineating the cause-and-effect relationship between the WSN-evoked hypothermia and reduced metabolism is important but challenging. In the present study, we used reduced metabolic parameters to characterize the torpor-like state. Future work is warranted to investigate the relationships between WSNs and metabolism.

We believe that DBS triggers the lowering of the central temperature control setpoint through the warm sensing neurons. This is a critical first step in reducing the T_{core}. The subsequent slowdown of metabolism may kick in to sustain the hypothermia and the torpor-like state. This is why we focused on the mechanisms of how DBS initiation of the lowering of T_{core}. Having said that, the hypothalamus comprises many different sub-areas involved in the regulation of temperature, metabolism, sleep, and so on, which is a complex system. The torpor induction and thermoregulatory pathways are closely intertwined, and hypothermia and hypometabolism may co-occur. It will be challenging and exciting work to distinguish body temperature and metabolism regulation in the hypothalamus in the future (PMID: 27813827: The Central Control of Energy Expenditure: Exploiting Torpor for Medical Applications; PMID: 35301430: Hypothalamic control of energy expenditure and thermogenesis; PMID: 22063719: Central control of thermogenesis). Only until then, the importance of the reduction in metabolism as a mechanism of DBS neuroprotection can be fully elucidated.

Nevertheless, to demonstrate that T_{core} is key for brain protection, we performed a new experiment to show that when DBS of MPNs was performed without hypothermia, the

ischemic brain was not protected, as shown in Fig. 4 (also attached below), suggesting that the decrease in Tcore is critical.

To provide more direct evidence to support WSN is the key to DBS-triggered hypothermia, we performed three new experiments as follows:

First, we performed detailed quantifications of the co-localization of c-Fos with BDNF and *Adcyap1* (mRNA) (and vice versa) using double immunohistochemistry and RNAscope *in situ* labelling, respectively. The co-localizations were shown on coronal sections cut at two Bregma points (AP: +0.15 and +0.26 mm) to indicate the c-Fos expressions in relation to the DBS electrode, which was at ML at 0.3 mm, AP at +0.15 mm, DV at 5.15 mm (Fig. 5a, c, as shown below). The highest intensity of c-Fos was at the medial preoptic area and medial preoptic nucleus (Fig. 5 h) and less in the lateral preoptic area (LPO). Based on these new data, we determined that the DBS site was truly in the MPN.

Second, we placed DBS electrodes at several places in the brain to determine the selectivity of DBS-induced hypothermia (Fig. 4g-k, shown below). DBS electrodes were placed bilaterally into LPO (ML0.8 mm), VP (ML1.2 mm) similar to the MPN site (ML0.3 mm) with AP at +0.15 mm, DV at 5.15 mm. Another DBS site used was VDB at ML0.3 mm, AP at +0.74 mm, and DV at 5.15 mm. Stimulation of these sites did not produce deep hypothermia, as seen in DBS MPN. Although there occurred shallow reductions of the core body temperature in VDB and LPO stimulations, the T_{core} did not drop below 35 °C. These data demonstrated that MPN is a specific target nucleus for DBS hypothermia.

Third, to unequivocally demonstrate that DBS activation of WSNs is key in inducing therapeutic level of hypothermia, inhibitory AAV2/9-hSyn-DIO-hM4D(Gi)-eGFP-WPRE-pA was injected into the MPN of *Adcyap1-2A-Cre* knock-in (B6.Cg-*Adcyap1^{tm1.1(cre)Hze/ZakJ}*) and *Vglut2-ires-cre* knock-in (C57BL/6J) mice to silence *Adcyap1* and *Vglut2* neurons, respectively. Silencing of excitatory *Vglut2* neurons completely blocked DBS-evoked hypothermia (Fig. 5m, shown below). Interestingly,

inhibiting the *Adcyap1* gene in the MPN did not completely block DBS-evoked reduction in Tcore (Fig. 5n), suggesting the involvement of other WSNs in the MPA in response to DBS- stimulation. These data demonstrated that DBS hypothermia requires the activation of WSNs.

Together, these experiments provided direct evidence showing that WSN in the MPN is the key to DBS-triggered hypothermia.

The manuscript could benefit from another round of editing to correct minor mistakes and improve clarity. For example, "chemogenetic" is spelled inconsistently.

Answer: Thank you, and we apologize for the mistakes. We have asked a native English-speaking colleague to proof-read the manuscript and correct the inconsistently spelled words such as "chemogenetic".

Reviewer #1 (Remarks to the Author):

The revisions have improved the manuscript. This is a nice paper.

Reviewer #2 (Remarks to the Author):

My concerns in the prior review were sufficiently addressed. I appreciate the additional experiments to proof the implant location.

Reviewer #3 (Remarks to the Author):

We thank the authors for their extensive and careful review, and for the additional experimentation included in the replies.

Reviewer #4 (Remarks to the Author):

the authors well-respond to comments from the reviewers. no further comments and suggestions.

Point-by-point Reply (answers in blue colour)

Reviewer #1 (Remarks to the Author):

The revisions have improved the manuscript. This is a nice paper.

Answer: We thank the reviewer for taking the time to provide helpful feedback.

Reviewer #2 (Remarks to the Author):

My concerns in the prior review were sufficiently addressed. I appreciate the additional experiments to proof the implant location.

Answer: We thank the reviewer for taking the time to provide helpful feedback.

Reviewer #3 (Remarks to the Author):

We thank the authors for their extensive and careful review, and for the additional experimentation included in the replies.

Answer: We thank the reviewer for taking the time to provide helpful feedback.

Reviewer #4 (Remarks to the Author):

the authors well-respond to comments from the reviewers. no further comments and suggestions.

Answer: We thank the reviewer for taking the time to provide helpful feedback.